# Combating subclonal evolution of resistant cancer phenotypes

Samuel W. Brady [1,2], Jasmine A. McQuerry[1,3], Yi Qiao[4], Stephen R. Piccolo [1,2,5], Gajendra Shrestha[1], David F. Jenkins [6], Ryan M. Layer[4], Brent S. Pedersen[4], Ryan H. Miller[3], Amanda Esch[7,8], Sara R. Selitsky[9], Joel S. Parker[9], Layla A. Anderson[1], Brian K. Dalley[10], Rachel E. Factor[11], Chakravarthy B. Reddy[12], Jonathan P. Boltax[12], Dean Y. Li[3,4], Philip J. Moos[1], Joe W. Gray [7,8], Laura M. Heiser[7,8], Saundra S. Buys[13], Adam L. Cohen[13], W. Evan Johnson[3,6], Aaron R. Quinlan[2,4], Gabor Marth[4], Theresa L. Werner[14] & Andrea H. Bild [1,2,3,15]

Metastatic breast cancer remains challenging to treat, and most patients ultimately progress on therapy. This acquired drug resistance is largely due to drug-refractory sub-populations (subclones) within heterogeneous tumors. Here, we track the genetic and phenotypic sub-clonal evolution of four breast cancers through years of treatment to better understand how breast cancers become drug-resistant. Recurrently appearing post-chemotherapy mutations are rare. However, bulk and single-cell RNA sequencing reveal acquisition of malignant phenotypes after treatment, including enhanced mesenchymal and growth factor signaling, which may promote drug resistance, and decreased antigen presentation and TNF-α signaling, which may enable immune system avoidance. Some of these phenotypes pre-exist in pre-treatment subclones that become dominant after chemotherapy, indicating selection for resistance phenotypes. Post-chemotherapy cancer cells are effectively treated with drugs targeting acquired phenotypes. These findings highlight cancer's ability to evolve phenotypically and suggest a phenotype-targeted treatment strategy that adapts to cancer as it evolves.

[1] Department of Pharmacology and Toxicology, College of Pharmacy, University of Utah, 30 South 2000 East, Salt Lake City, UT 84112, USA. [2] Department of Biomedical Informatics, School of Medicine, University of Utah, 421 Wakara Way, Salt Lake City, UT 84108, USA. [3] Department of Oncological Sciences, School of Medicine, University of Utah, 2000 Circle of Hope Drive, Salt Lake City, UT 84112, USA. [4] Department of Human Genetics, School of Medicine, University of Utah, 15 South 2030 East, Salt Lake City, UT 84112, USA. [5] Department of Biology, College of Life Sciences, Brigham Young University, Provo, UT 84602, USA. [6] Division of Computational Biomedicine, School of Medicine, Boston University, 72 East Concord Street, Boston, MA 02218, USA. [7] Department of Biomedical Engineering, Oregon Health & Science University, 2730 SW Moody Ave, Portland, OR 97201, USA. [8] Oregon Center for Spatial Systems Biomedicine, Oregon Health & Science University, 2730 SW Moody Ave, Portland, OR 97201, USA. [9] Department of Genetics and Lineberger Comprehensive Cancer Center, University of North Carolina, 450 West Drive, Chapel Hill, NC 27599, USA. [10] High-Throughput Genomics and Bioinformatic Analysis, Huntsman Cancer Institute, 2000 Circle of Hope Drive, Salt Lake City, UT 84112, USA. [11] Department of Pathology, Huntsman Cancer Hospital, 1950 Circle of Hope Drive, Salt Lake City, UT 84112, USA. [12] Department of Internal Medicine, Pulmonary Division, School of Medicine, University of Utah, 26 North Medical Drive, Salt Lake City, UT 84132, USA. [13] Department of Internal Medicine, Huntsman Cancer Institute, University of Utah, 2000 Circle of Hope Drive, Salt Lake City, UT 84112, USA. [14] Department of Medicine, Oncology Division, Huntsman Cancer Institute, University of Utah, 2000 Circle of Hope Drive, Salt Lake City, UT 84112, USA. [15] Department of Medical Oncology and Therapeutics, City of Hope Comprehensive Cancer Institute, 1218 S Fifth Ave, Monrovia, CA 91016, USA. Correspondence and requests for materials should be addressed to A.H.B. (email: andreab@genetics.utah.edu)

Each patient's tumor has the potential for a unique evolutionary trajectory. Tumor subclones, defined as cells with distinct genetic lineages, have revealed remarkable genomic heterogeneity in most epithelial cancers, providing a substrate for evolution under the selective pressure of treatment[1, 2]. Solid tumors generally lack significant numbers of common actionable mutations, making it difficult to link mutational genotype to an obvious treatment strategy[3, 4]. In addition, tumor cell phenotypes, defined by processes such as cell growth, survival, and differentiation states, can also evolve over time due to genetic, epigenetic, or environmental factors[5, 6]. Our approach focuses on linking these two phenomena—clonal evolution and genomic diversity—by tracking changes in subclonal structure over time to identify and target phenotypes driving drug resistance that emerge as tumors progress. As the majority of genetic alterations found in resistant tumor subclones occur in a small proportion of tumors and do not lead to survival advantage[7, 8], characterizing patient tumors by these more generalizable oncogenic phenotypes can facilitate directed drug treatment.

Our current study focuses on the metastatic setting, where cancer is usually not curable. Currently, treatment decisions are based on the availability of targeted therapies (for HER2+ and ER + cancers) and on metastatic site, symptoms, prior use of chemotherapy, and overall health, and comorbidities[9]. Therefore, treatment decisions are generally made independent of patient tumor phenotype or heterogeneity and do not account for temporal cancer evolution[10].

Here, we use DNA sequencing data from four breast cancer patients, followed for years, to delineate the genetic events occurring in cancer cells as they change during treatment with different drugs, and to identify the cancer's subclonal evolution in response to therapy. Further, bulk and single-cell RNA sequencing data identify gene expression patterns, or signatures, for key pathways that represent specific cellular phenotypes, such as cell growth and death processes. Critically, these data are used to link tumor subclone evolution to emerging oncogenic phenotypes associated with acquired resistance. We develop treatment strategies that target phenotypes in resistant tumor subclones that are polyclonal and/or phenotypically unique. Altogether, our research provides genomic assessment of tumor subclones combined with a dynamic approach that could allow adaptive therapy that matches the tumor's capacity for evolution.

## Results

**Patient treatment history and approach.** Genetic and phenotypic evolution of four metastatic ER+ breast cancers was examined over 2–15 years and 3–6 samples per patient. Patients were selected based on the availability of repeated longitudinal samples, generally from metastatic pleural or ascites fluids. For each patient, subclonal evolution was identified through bulk and/or single-cell DNA sequencing at multiple points in the patient's treatment history (Fig. 1, #1 and #2). RNA-Seq identified biological phenotypes associated with these evolving subclones, and effective treatments for post-chemotherapy subclones, as shown by drug assays using patient tumor cells (Fig. 1; #3 and #4).

**Subclonal heterogeneity and evolution of four breast cancers.** Subclonal evolution of four breast cancers was determined with 60 × whole-genome sequencing (WGS), 100 × whole-exome sequencing (WES) and targeted single-cell DNA sequencing, along with SubcloneSeeker[11] analysis. Variants identified were validated by detection in RNA-Seq data (Supplementary Fig. 1), single-nucleotide polymorphism (SNP) array (Supplementary Fig. 2), and matched clinical sequencing results for *BRCA2*-mutant patients. The specific sequencing performed for each patient is indicated by filled squares (WGS), inverted triangles (WES), and empty squares (scDNA-Seq; Fig. 2). Somatic WGS single-nucleotide variants (SNVs) and indels found in copy-neutral regions were grouped into mutation clusters based on their presence or absence at specific timepoints. Next, the cancer cell fraction (CCF—the percent of cancer cells harboring a mutation cluster) of each mutation cluster was determined by identifying the variant allele frequency (VAF) of maximum density at each timepoint; subclones were then classified using rules previously described[11]. For patient #1, this approach was validated by SubcloneSeeker analysis of WES data (Supplementary Fig. 3) and refined by targeted single-cell DNA sequencing (scDNA-Seq). scDNA-Seq was performed by PCR-amplifying regions surrounding 17 subclonal mutations, identified from WES, from single-cell DNA prepared using Fluidigm C1, followed by next-generation sequencing, which confirmed SubcloneSeeker predictions (Supplementary Fig. 4, Methods section). Subclones are represented as the outer circles containing mutation clusters (represented by smaller colored circles) in Fig. 2, with the CCF of each subclone indicated as a percentage next to each subclone. In addition, VAFs for copy-neutral somatic mutations (SNVs and indels) used to determine subclones are shown as a heatmap to the left of each panel in Fig. 2. Copy-number alterations (CNAs) and structural variants for each patient are shown in Circos plots in Fig. 3. Mutations in Cancer Gene Census[12] "cancer genes" are shown explicitly, by name, in Figs. 2 and 3.

Patient #1 (ER + /HER2 + ) samples were collected over more than 2 years from malignant pleural effusion and ascites fluids. Samples were obtained before, during, or after the following treatments: (1) paclitaxel and trastuzumab followed by docetaxel, (2) liposomal doxorubicin, (3) trastuzumab and MM-111 (an experimental HER2/HER3 antagonist[13]) followed by (4) carboplatin and gemcitabine (Fig. 2a; see Supplementary Data 1 for full history), after which the patient succumbed to disease. Truncal mutations (defined as mutations present in all subclones and timepoints) in patient #1 included loss of heterozygosity of the

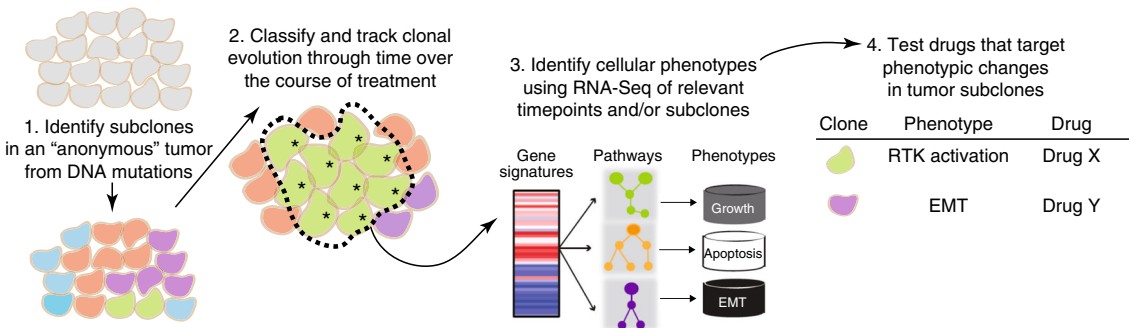

**Fig. 1** Overview of systems approach for identifying therapeutic vulnerabilities from longitudinal genomic analysis. *resistant subclone

patient's germline *BRCA2* E1493fs mutation, an *ESR1* L538P (activating[14]) mutation, homozygous structural variants (likely inactivating) in *SMAD4* and *MAP2K4* (Figs. 2a and 3a), and increased *ERBB2* copy (3 copies), consistent with HER2 + status (Supplementary Fig. 5). Following a response to paclitaxel with

trastuzumab, the patient acquired three new subclones, suggesting independent acquired resistance mechanisms ("Tax + trast"; see Fig. 2a). One of these subclones, SC2, appeared at low CCF after paclitaxel and trastuzumab (<1%) but came to dominate with CCF of 100% after subsequent treatment with liposomal

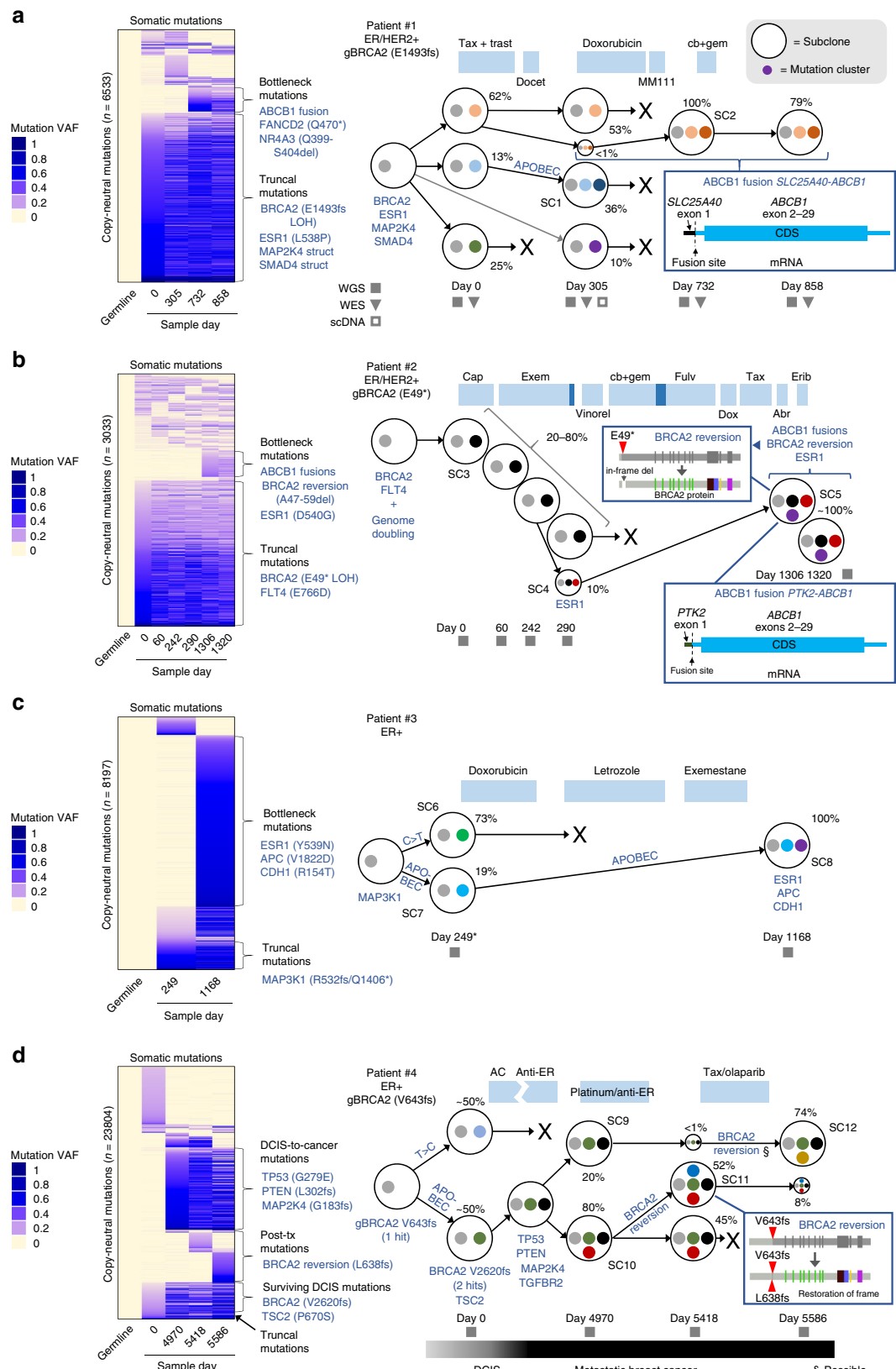

doxorubicin ("Doxorubicin"; Fig. 2a), to which the patient partially responded. This bottleneck subclone possessed an *SLC25A40-ABCB1* fusion resulting from a structural deletion on chromosome 7, which fused the active promoter and 5′ UTR of *SLC25A40* to the *ABCB1* gene, preserving the entire *ABCB1* coding region and leading to increased levels of the ABCB1[15] drug efflux pump (Fig. 2a, inset and Supplementary Figs. 6 and 7a). SC2 also possessed a copy gain on chromosome 10, including *CDK1*, and loss on chromosome 2 (Fig. 3a, blue arrows). In addition, the bottleneck subclone SC2 possessed 1047 new SNVs and indels ("Bottleneck mutations" in Fig. 2a, left; dark-orange circles in Fig. 2a, right), including mutations in cancer-associated genes *NR4A3* and *FANCD2*[12].

Previous research shows that summarizing mutations by trinucleotide context reveals tumor-specific mutation signatures[16]. To determine how mutation signatures evolve over time, we first ascertained the trinucleotide context of truncal and subclone-specific SNVs (Fig. 4a). We then quantified the presence of 30 mutation signatures[16] from COSMIC[12] in various subclones[17] (Fig. 4b). Interestingly, both truncal and subclonal variants showed enrichment of BRCA-deficiency-induced mutations in patient #1 (Fig. 4b, signature 3), consistent with *BRCA2* inactivation. In contrast, APOBEC (apolipoprotein B mRNA editing enzyme, catalytic)-associated mutations[16] evolved significantly, appearing in a subclone after paclitaxel and trastuzumab (subclone SC1; Figs. 2a, 4a, third column, and Fig. 4b, third bar). Indeed, 31.3% of subclone-specific SNVs in this subclone were APOBEC-associated (Fig. 4a, asterisks) compared to 10.0% of subclone-specific variants in its parental subclone (Fig. 4a, second column and Fig. 4b, second bar; $P = 5.5 \times 10^{-22}$ by two-sample proportion test), and 12.7% of truncal (gray) variants (Fig. 4a, first column and Fig. 4b, first bar; $P = 1.5 \times 10^{-53}$ by two-sample proportion test). As a negative control, germline SNPs in patient #1 (and other patients) were also analyzed; these lacked the BRCA-deficiency and APOBEC signatures (Supplementary Fig. 8). Together, these data indicate that some mutational processes can evolve and differ (APOBEC) between subclones, while others (BRCA-associated) remain constant during cancer progression.

Patient #2 (ER + /HER2 + ) was followed for over 3 years, including 6 pleural effusions. Samples were obtained before, during, or after the following treatments: (1) capecitabine, (2) exemestane, (3) exemestane and everolimus, (4) vinorelbine, (5) carboplatin and gemcitabine, (6) fulvestrant, (7) liposomal doxorubicin, (8) paclitaxel, (9) Abraxane, and (10) eribulin (Fig. 2b, Supplementary Data 2). This patient was pseudo-tetraploid, as determined by CNAs inferred from WGS (Supplementary Fig. 9a–c) and DNA content quantification (Supplementary Fig. 9d), due to an early genome doubling event, necessitating modifications to subclone identification (Methods section). Patient #2 possessed a germline heterozygous *BRCA2* E49* mutation that underwent loss of heterozygosity in the patient's cancer, a possibly oncogenic *FLT4* E766D somatic mutation, and moderate *ERBB2* amplification (5 copies; Supplementary Fig. 5) at all timepoints. This patient initially possessed one major subclone (SC3) with a CCF of <80% (Fig. 2b).

Towards the end of treatment with the anti-estrogen exemestane a new minor subclone (SC4), with CCF of 10%, appeared possessing an *ESR1* D540G mutation (Figs. 2b and 3b), which may have promoted exemestane resistance[14]. The patient's disease progressed, and after more than two years and numerous treatments, patient #2's cancer was dominated by a new subclone (SC5), derived from the *ESR1*-mutant subclone SC4, with CCF of ~100% (Fig. 2b). The bottleneck subclone SC5 possessed 936 new SNVs and indels ("Bottleneck mutations"; Fig. 2b heatmap). One of these, an in-frame *BRCA2* A47-P59 deletion, removed the inactivating frameshift in one copy of *BRCA2* to likely restore function and gain resistance to platinum therapy[18] (Fig. 2b, "BRCA2 reversion" inset, and Supplementary Fig. 10a). Additionally, SC5 acquired two unique *ABCB1* fusions (*PTK2-ABCB1* and *AFF3-ABCB1*) that also provided *ABCB1* with a strong promoter while maintaining the *ABCB1* coding region intact (Fig. 2b, bottom-right inset and Supplementary Fig. 11), and apparently promoted *ABCB1* expression (Supplementary Fig. 7b). It is unclear whether the two fusions were acquired sequentially due to selective pressure for additional *ABCB1* expression, or co-existed in the original refractory subclone. Unlike patient #1, new mutational signatures did not appear in patient #2 after treatment, and the BRCA-deficiency signature was relatively constant, consistent with *BRCA2* inactivation (Fig. 4a, b). The BRCA-deficiency signature in *BRCA2*-revertant SC5 is potentially due to BRCA-loss-induced mutagenesis in an SC5 precursor prior to the reversion event.

Patient #3 (ER + /HER2-) was followed for more than 3 years, including 3 pleural effusions. Samples were obtained before, during, or after: (1) two doxorubicin courses, (2) letrozole, and (3) exemestane (Fig. 2c, Supplementary Data 3). The patient showed clinical benefit from each treatment followed by progression. Truncal mutations included *MAP3K1* mutations (R532fs and Q1406*) inactivating this tumor suppressor[19]. Initially two subclones existed: SC6, possessing mostly C > T mutations (Fig. 4a), dominated with a CCF of 73%, while subclone SC7 represented 19% of cells (Fig. 2c). However, after doxorubicin, letrozole, and exemestane a new SC7-derived subclone came to dominate with a CCF of 100% (SC8; Fig. 2c). This bottleneck subclone possessed a striking 5540 additional SNVs and indels (Fig. 2c, "Bottleneck mutations"), representing six times the number truncal SNVs ($n = 921$). 67.4% of the new bottleneck SNVs were APOBEC-associated compared to 11.1% of truncal SNVs (Fig. 2c; Fig. 4a, see "SC8"; $P = 9.8 \times 10^{-275}$ by two-sample proportion test). APOBEC signature weights were likewise higher among bottleneck mutations (Fig. 4b, see "SC8" and signatures 2/13) compared to truncal, and pre-existed in the bottleneck subclone's precursor before treatment (Fig. 4b, "SC7"), suggesting a genetic lineage with enriched APOBEC mutagenesis. Among the 5540 new mutations in SC8 were *ESR1* Y539N, which may have promoted letrozole or exemestane resistance[14], *APC* V1822D, and *CDH1* (E-cadherin) R154T mutations (Figs. 2c and 3c). Additionally, the bottleneck subclone SC8 gained copies of chromosome 5p, and lost regions of 1p, 2q, 15, and 22 (Fig. 3c, green arrows). Lost regions included apoptosis genes *CASP8*, *CASP10*, and *BID*, potentially leading to apoptosis defects.

**Fig. 2** Subclonal evolution of four breast cancers over 2–15 years. **a–d** Subclonal evolution of breast cancer patients #1 to #4 through treatment. Left side shows variant allele frequencies of copy-neutral somatic SNVs and indels (WGS) organized into clusters, with relevant cancer-associated mutations (may or may not be copy-neutral and includes structural variants) indicated. Right shows subclone evolution. Subclones are indicated by large circles; mutation clusters are indicated by small colored circles. Relevant mutations in subclones are indicated by text or boxed insets. CCF is indicated as percent next to subclone. Filled squares indicate timepoints sequenced by whole-genome sequencing (WGS), filled inverted triangles indicate whole-exome sequencing (WES), and empty squares indicate targeted single-cell DNA sequencing (scDNA). Abr, Abraxane; cap, capecitabine; cb, carboplatin; dox, doxorubicin; erib, eribulin; exem, exemestane; fulv, fulvestrant; gem, gemcitabine; Tax, paclitaxel; trast, trastuzumab; vinorel, vinorelbine. *Patient #3 day 0 had RNA-Seq only. gBRCA2 indicates a germline *BRCA2* mutation

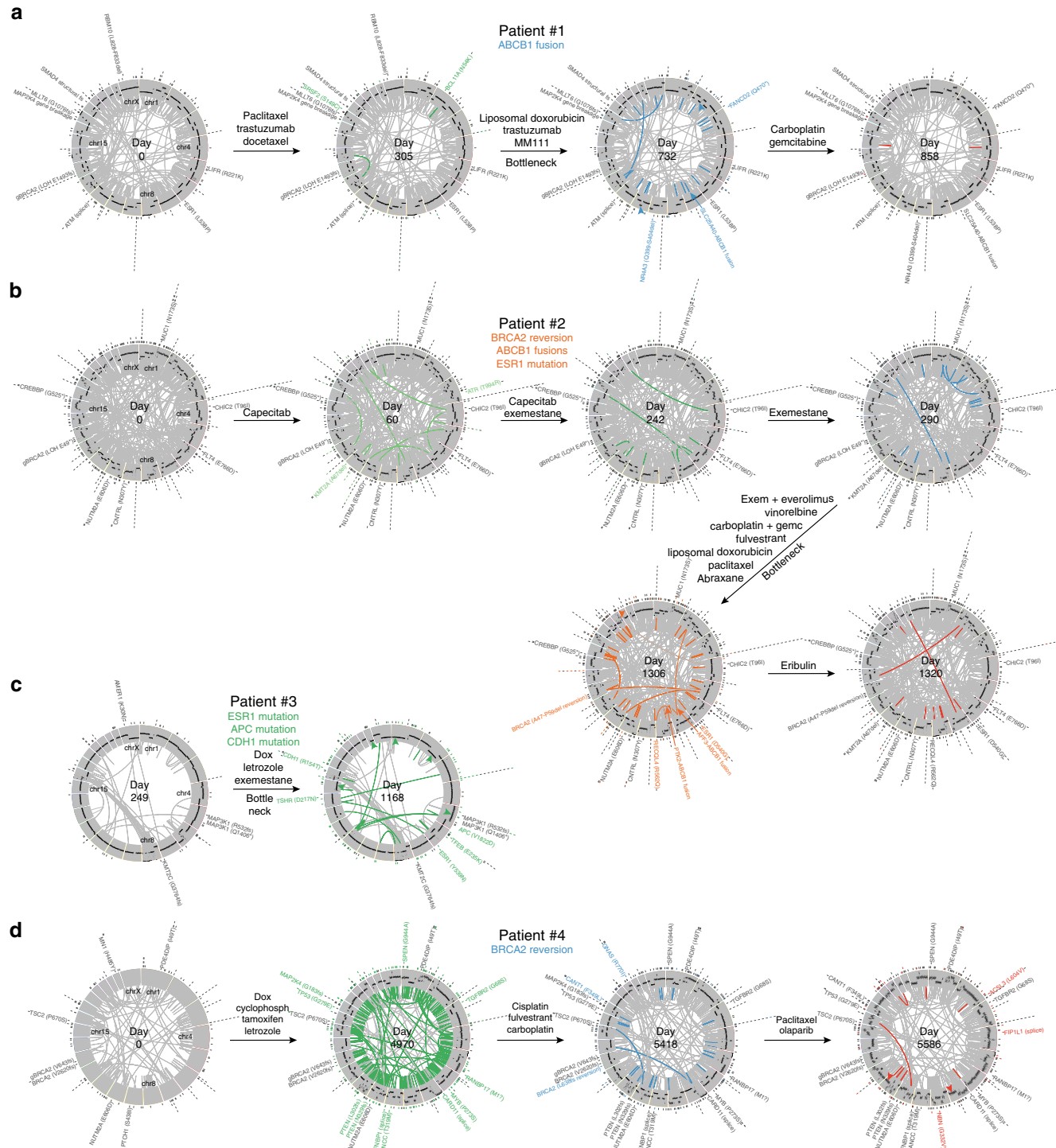

**Fig. 3** SNV, structural variant, and CNA evolution in four breast cancers. **a–d** Circos plots showing mutation evolution of breast cancer patients #1 to #4 through treatment. Each circle represents chromosomes 1 through X (clockwise from top) arranged in a circle. Protein-coding somatic SNVs and indels are indicated outside each circle as ticks or, for Cancer Gene Census genes, by name. Germline *BRCA2* mutations are indicated as "gBRCA2." Structural variants (large deletions, translocations, inversions, and duplications) are indicated inside each circle as a line joining the start and end of the variant, with Cancer Gene Census gene mutations indicated outside. Copy number changes are represented in the gray region with higher copy towards the outer edge. Newly appearing SNVs, indels, and structural variants are shown in color, while selected newly appearing CNAs are indicated by colored arrows. CNAs were not determined in the patient #4 day 0 sample due to technical issues with FFPE samples

Patient #4 (ER + /HER2-) was followed for 15 years, including her primary tumor ductal carcinoma *in situ* (DCIS) and recurrent pleural effusions, including the following treatments: (1) doxorubicin with cyclophosphamide ("AC"; Fig. 2d), (2) anti-estrogens tamoxifen and letrozole, with capecitabine between, (3) cisplatin followed by fulvestrant, interferon treatment for a non-cancer diagnosis, carboplatin, then tamoxifen ("Platinum/anti-ER"), and (4) paclitaxel followed by olaparib ("Tax/olaparib"; Fig. 2d, Supplementary Data 4). This patient possessed a germline *BRCA2* inactivating (V643fs) mutation, and relatively few truncal

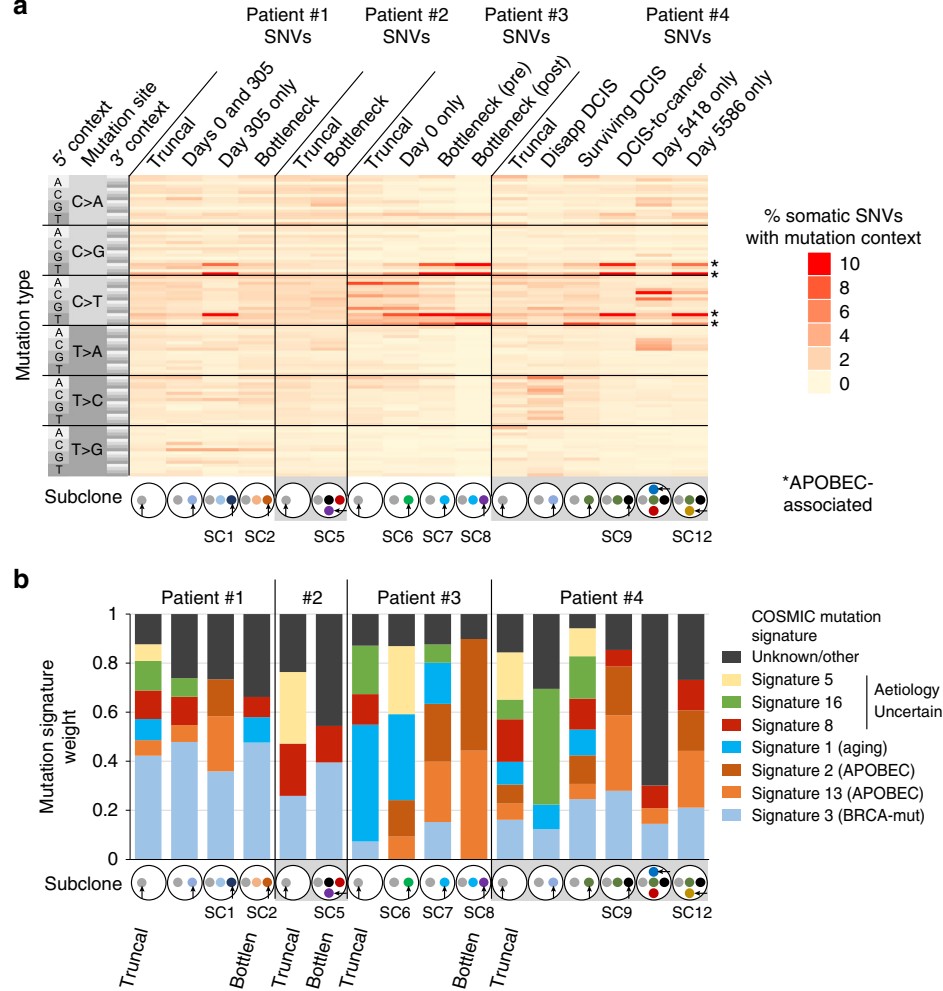

**Fig. 4** Mutation signature evolution in four breast cancers. **a** Heatmap showing percent of SNVs in indicated subclones, defined in Fig. 2, with each of 96 possible mutation/trinucleotide context combinations. **b** Mutation signature weights for COSMIC mutation signatures in indicated subclones. Signatures with <0.03 average weight and mutations not accounted for by known signatures are included in the unknown/other group

mutations. Two independent tumor subclones were present at a CCF of ~50% each at day 0, with a *BRCA2* V2620fs mutation, representing the surviving subclone, which also possessed a *TSC2* P670S mutation, giving rise to subsequent samples (Figs. 2d and 3d). (These two subclones were likely independent since the survivor (dark-green) had a minor peak possibly representing a surviving subclone within it, precluding the possibility of blue mutations being in the same cells as dark-green based on VAF densities and opposing evolutionary pattern of the two clusters (Supplementary Fig. 12). 13 years later, after transition from DCIS to metastatic disease, the patient had acquired additional mutations in *TP53* (G279E homozygous), *PTEN* (L302fs and N329fs), *MAP2K4* (G183fs homozygous), and *TGFBR2* (G68S). At this first metastatic timepoint (day 4970) there were two major subclones, SC9 (CCF of 20%) and SC10 (80%). Subsequently the patient received platinum-based therapy (cisplatin and carboplatin) which selected for SC11 (evolved from SC10). SC11 contained a *BRCA2* reversion (L638fs) restoring the frame of the V643fs mutation after only a few aberrant amino acids, likely restoring BRCA2 function (Fig. 2d, bottom-right inset and Supplementary Fig. 10b). The *BRCA2* V643fs germline mutation and somatic reversion (L638fs) were in *cis* as seen in reads spanning both mutations (Supplementary Fig. 13). After further treatment with paclitaxel and olaparib, the BRCA2-revertant SC11 decreased from CCF of 52% to 8%, while SC12 became

dominant at 74%. SC12, which survived olaparib, may have acquired a direct reversion[20] of the germline *BRCA2* V643fs mutation, as the V643fs VAF decreased to 0.28 at day 5586 from an average of 0.51 in all other samples, including germline (Supplementary Fig. 14). *BRCA2* reversions are thought to also promote olaparib resistance[21]. Interestingly, even though both SC11 and (possibly) SC12 may have restored BRCA2 function, the response to paclitaxel and olaparib was different between the two subclones, with one responding and one non-responsive, suggesting additional resistance mechanisms were important for survival to these drugs. In addition, patient #4's DCIS-to-metastatic conversion was accompanied by enrichment in APOBEC-associated mutations (Fig. 4a, b, see "SC9") compared to truncal mutations (40.9% APOBEC-associated mutations among post-DCIS SNVs vs. 16.6% in truncal; $P = 2.0 \times 10^{-34}$ by two-sample proportion test), while SC12 was also enriched in APOBEC mutations compared to truncal (32.7%; $P = 8.7 \times 10^{-17}$ by two-sample proportion test).

While each patient's subclone evolution differed, several themes emerged. First, effective treatments resulting in long-term cancer control generally resulted in genetic bottleneck events in which one major subclone survived (SC2, SC5, and SC8). Second, two patients acquired *ABCB1* promoter fusions previously unreported in breast cancer, potentially promoting drug resistance. Third, *BRCA2* reversion events in breast cancer

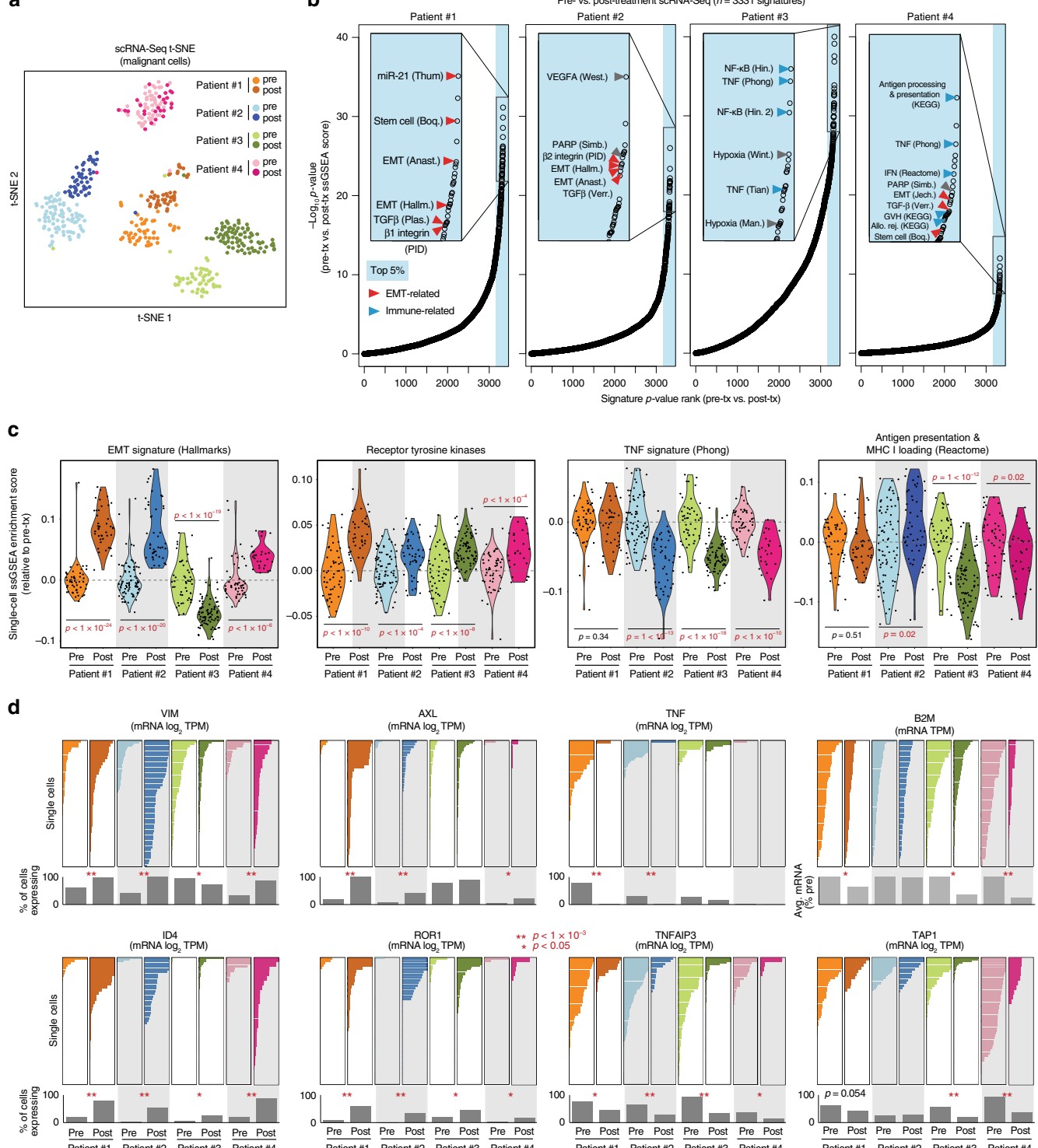

**Fig. 5** Single-cell RNA-Seq of pre- and post-treatment breast cancers reveals increased immune-avoidance, EMT, and RTK phenotypes. **a** t-SNE analysis of 428 individual cells' expression profiles from pre- and post-treatment breast cancer samples from four patients. **b** Plots of $P$-values ($t$-test-derived) comparing ssGSEA enrichment scores for 3331 C2 signatures between pre- and post-treatment single cells in each patient. $x$-axis, $P$-value ranks (higher ranks are more significant); $y$-axis, $-\log_{10}(P\text{-value})$. Red arrows, EMT- and stem cell-related signatures; blue arrows, immune-related signatures. **c** ssGSEA enrichment score (relative to pre-treatment average) violin plots for single cells in each sample for indicated signatures. Each point represents a single cell. $P$-values are by Student's $t$-test (two-tailed). **d** Expression of indicated genes by scRNA-Seq ($x$-axis) in individual cells ($y$-axis) with percent of cells expressing each gene or average expression indicated below in gray bars. $P$-values are by two-sample proportion test. *$P < 0.05$. **$P < 1 \times 10^{-3}$. $x$-axis scales all begin at zero (left) and are the same within patients for each gene

may contribute to resistance in platinum and/or PARP inhibitor-treated *BRCA2* carriers. Lastly, mutational signatures differed significantly between subclones. Some mutational signatures (APOBEC-associated) can apparently be acquired *de novo*, independent of the parent subclone's mutational processes, while others, such as BRCA-loss-associated signatures, are more consistent throughout evolution (Fig. 4). This is consistent with previous findings that APOBEC-associated mutagenesis appears later during lung cancer progression[22]. Acquisition of new mutational processes may provide a fitness advantage to some subclones, as their ability to evolve in response to treatment may be enhanced if a new mutational process provides bursts of mutagenesis[23] increasing genetic diversity.

**Single-cell RNA-Seq identifies resistance phenotypes.** Single-cell RNA-Seq (scRNA-Seq) has several advantages over bulk RNA-Seq, including analysis of intratumoral heterogeneity and exclusion of non-malignant cells affecting bulk measurements[24]. We performed scRNA-Seq on all four patients, at two timepoints each which were reflective of either pre- and post-bottleneck event (patients #1-3) or first metastatic timepoint pre-treatment ("sensitive") and the last timepoint following progression ("resistant"). Specific timepoints measured were: patient #1 days 305 and 732, patient #2 days 290 and 1320, patient #3 days 249 and 1168, and patient #4 days 4970 and 5586. We identified cells as normal or cancer based on their copy number profiles inferred from scRNA-Seq[24] (Supplementary Figs. 15–18). As expected, we found that non-copy-altered (normal) cells expressed higher mesothelial and fibroblast markers than copy-altered (cancer) cells[25, 26] (Supplementary Fig. 19). This led to a dataset of 428 individual breast cancer cells from 8 specimens generated using the Fluidigm C1 scRNA-Seq platform, which currently has among the highest sensitivity for detecting lower-abundance RNA molecules, high quantification accuracy[27], and enables microscopic visualization to exclude doublets. We detected a median of 4739 genes per cell (Supplementary Fig. 20a), comparable to previous studies[24], and found concordance between matched bulk RNA-Seq and scRNA-Seq averaged across cells simulating a bulk sample ($r = 0.90$ with ~18,000 genes detected in both bulk RNA-Seq and scRNA-Seq simulated bulk sample, Supplementary Fig. 20b), indicating good data quality.

To identify phenotypes emergent in resistant tumors, we first used t-SNE analysis to observe global transcriptional differences in resistant versus sensitive tumor cells (Fig. 5a). We then identified pathway signatures changing between pre- and post-treatment samples, rather than individual genes, since multi-gene signatures are more stable against technical noise and gene dropout in scRNA-Seq data[28] (see schematic in Supplementary Fig. 20c). We applied the 3331 C2 signatures[29] to each single cell using ssGSEA[30] to obtain enrichment scores for each signature, and ranked each signature by its *P*-value comparing pre- and post-treatment single-cell enrichment scores by *t*-test in each patient (Fig. 5b). This analysis revealed that two broad classes of signatures were most dramatically different between pre- and post-treatment samples: (1) epithelial-mesenchymal transition (EMT)- and stem-cell-associated[31] signatures (red arrows), and (2) immune-associated signatures (blue arrows), including tumor necrosis factor alpha (TNF-α) and antigen presentation signatures. Specifically, we observed increased EMT in 3 patients, suggesting that EMT may promote chemoresistance (Fig. 5c). Further, our analysis revealed that TNF-α signaling and antigen presentation decreased after treatment in multiple patients (Fig. 5c). This suggests that treatment selects for subclones that avoid immune responses, as loss of both TNF-α signaling[32] and antigen presentation[33] are likely to suppress basal or treatment-

induced immune responses. Due to their importance and targetability in cancer, we also analyzed signatures for receptor tyrosine kinase activation[34], including the downstream Akt and K-Ras pathways. A signature inclusive of all 58 receptor tyrosine kinases (RTKs)[34] showed an increased post-treatment RTK phenotype in multiple patients (Fig. 5c). Further, our custom experimentally generated Akt and K-Ras signatures[35] showed that the Akt signature was increased in patients #2 and #3 and the K-Ras signature was increased in patients #1, #3, and #4 (Supplementary Fig. 21), indicating that increased post-treatment RTK expression may have promoted increased Akt and Ras signaling. We validated the significance of pathway dysregulation using PAGODA, which addresses technical noise in scRNA-Seq experiments using a probabilistic over-dispersion-based approach[28] (Supplementary Fig. 22). To test whether these results may be due to batch effects rather than biological differences, we ran the patient #2 pre-treatment (day 290) sample on two different capture chips (different days), and found that EMT, RTK, and TNF-α phenotypes were consistent between these two pre-treatment replicates compared to post-treatment (Supplementary Fig. 23a). Expression was highly concordant between replicates ($r = 0.91$; Supplementary Fig. 23b). Further, the upregulation of RTK pathways, EMT state, and anti-apoptotic signaling as resistance developed in patients were also seen in bulk RNA-Seq (Supplementary Figs. 24a, 25 and 26) and western blot and immunofluorescence using patient cells (Supplementary Figs. 24b, c and 27).

Individual genes related to the phenotypes were statistically analyzed using the proportion of tumor cells expressing each gene between pre- and post-treatment samples in each patient (Supplementary Fig. 28). This revealed significant differences, by two-sample proportion test ($P < 0.05$), between pre- and post-treatment proportions of cells expressing the EMT- and stem cell-associated genes *VIM* (vimentin[31]) and *ID4*[36]; the RTK genes *AXL* and *ROR1*[34]; the TNF-α pathway genes *TNF* (TNF-α) and *TNFAIP3*[37]; and the antigen processing gene *TAP1*[38] (Fig. 5d). In addition, the essential antigen presentation gene *B2M*, encoding the β2-microglobulin subunit required for HLA class I cell-surface expression[39], was expressed in all cells but was statistically decreased ($P < 0.05$ by *t*-test) in 3 of 4 patients (Fig. 5d, top-right).

**Resistance phenotypes pre-exist subclonally.** Our analysis shows that cancers acquire additional malignant phenotypes in response to therapy. This likely occurs through (1) genetic selection for subclones with these features and/or (2) drug-induced changes. While the studies above identify phenotypes emergent across all tumor cells in resistant states, we next wanted to test whether acquired phenotypes were present before treatment in survivor subclones (scenario #1). Thus, using the subclonal characterization based on the DNA sequencing analysis described previously, we interrogated if the subclones that survive chemotherapy exhibited key phenotypes prior to treatment, or if these phenotypes emerged only after treatment. To assign each cell to a subclone, we primarily used CNAs inferred from scRNA-Seq[24] data (Supplementary Figs. 15–18; "*" indicates relevant CNAs). A window size of 101 genes was used for averaging gene expression to infer CNAs from scRNA-Seq[24], which gave superior noise reduction relative to smaller windows and similar results to larger windows (Supplementary Fig. 29). CNAs that were partially present before treatment, and subsequently increased post-treatment (i.e., 2.5 then later 3.0 copies), were assumed to represent a pre-treatment survivor subclone, while those that were partially present pre-treatment but later disappeared (i.e., 2.5 then later 2.0 copies) were assumed to represent a

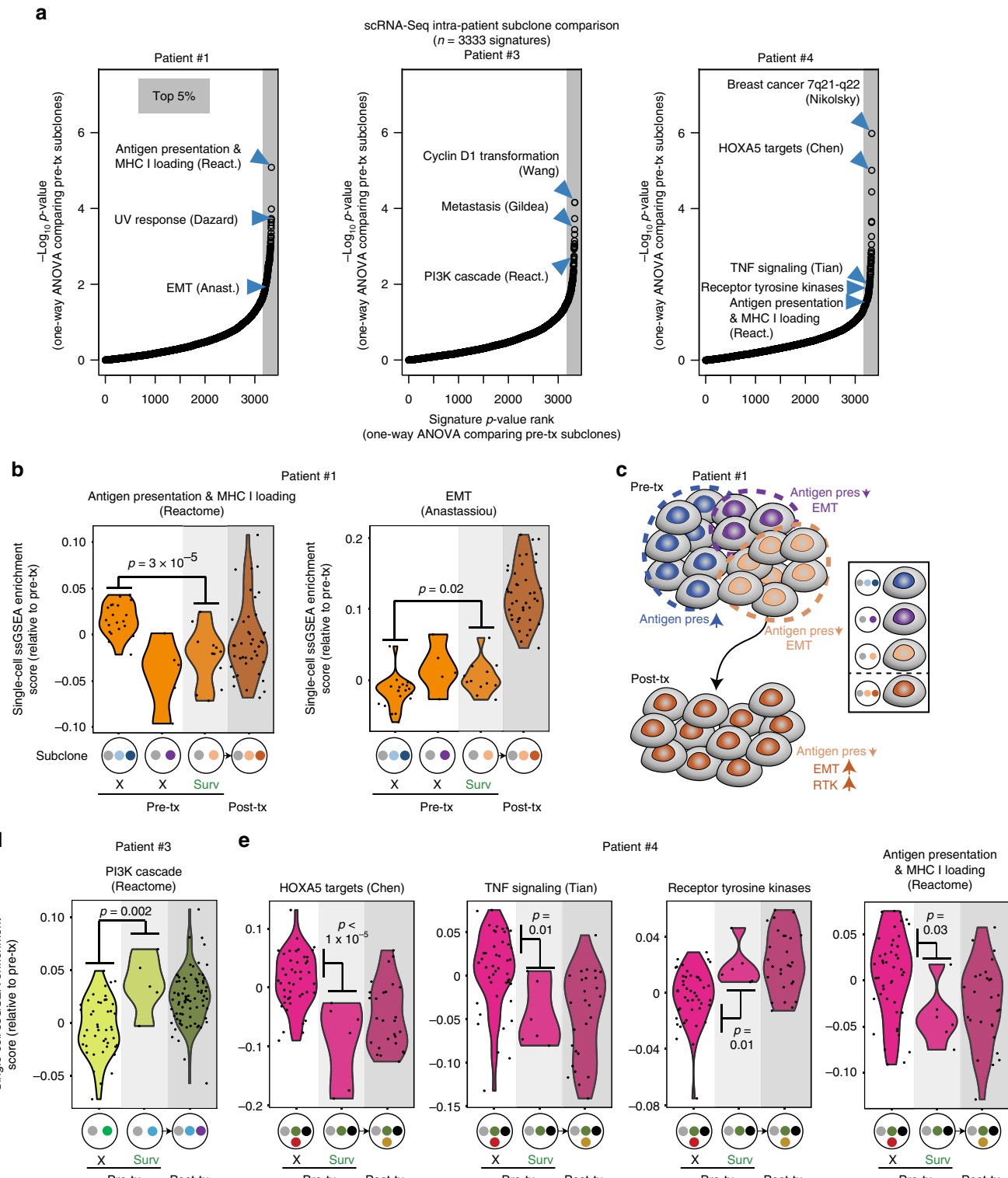

**Fig. 6** Pre-existence of post-treatment phenotypes in pre-treatment survivor subclones. **a** Plots of *P*-values (one-way ANOVA) comparing scRNA-Seq ssGSEA enrichment scores for 3331 C2 signatures, plus the receptor tyrosine kinases (all-58) signature and our anti-apoptosis signature, between subclones in each indicated patient's pre-treatment sample. *x*-axis, *P*-value ranks (higher ranks are more significant); *y*-axis, −log₁₀(*P*-value). **b** Patient #1 ssGSEA enrichment scores for single cells in each pre-treatment subclone, with post-treatment cells shown for comparison and the dominant post-treatment subclone indicated. Each dot represents a single cell. *P*-values are by Student's *t*-test. Subclones correspond to those shown in Fig. 2a. "X" indicates disappearing subclone while "Surv" indicates the subclone giving rise to the post-treatment sample. **c** Schematic showing subclonal phenotypic heterogeneity and evolution in pre-treatment patient #1 cells. **d** As in **b** but for patient #3. Subclones correspond to those in Fig. 2c. **e** as in **b** but for patient #4, with subclones as shown in Fig. 2d

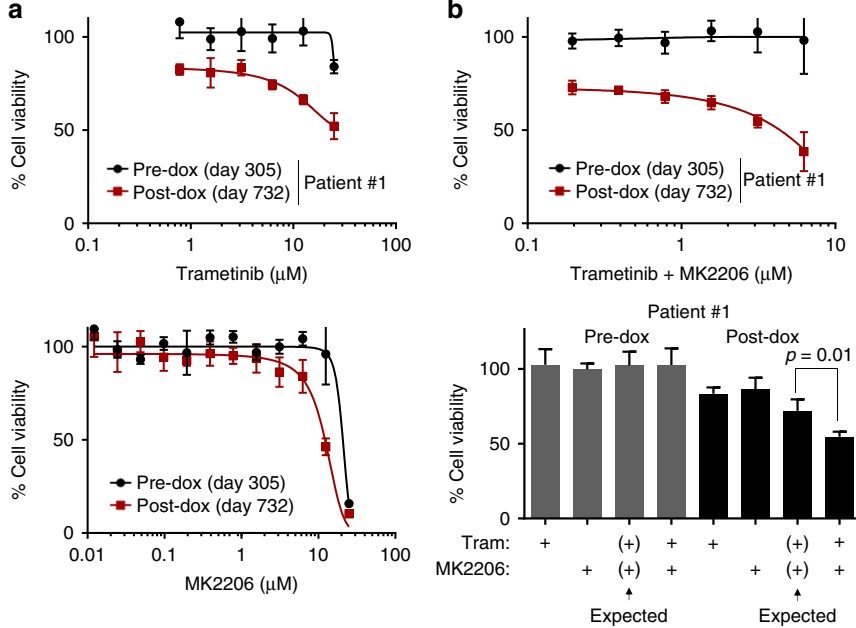

**Fig. 7** Acquired sensitivity to drugs targeting post-chemotherapy phenotypes. **a** Drug response assay comparing pre- and post-doxorubicin patient #1 cancer cells' sensitivity to drugs after 3-day treatment (fibroblast feeder system; CellTiter-Glo was used). **b** Indicated cells were treated with equimolar doses of trametinib and MK2206 for 3 days, followed by CellTiter-Glo (top); synergy analysis of 3.125 μM doses of these drugs (bottom). Expected is by Bliss independence; P-value is by Student's t-test (two-tailed). Fibroblast feeder system was used and fibroblast signal was subtracted out in **a**, **b** and percentages are relative to DMSO control mean. Error bars show s.d. of four technical replicates

disappearing subclone. See Supplementary Fig. 30 for identification of such CNAs, which were insufficiently detectable in patient #2, who was omitted from this analysis. Single cells were assigned to either survivor or disappearing subclones based on these CNAs (Supplementary Figs. 15–18). Subclonal CNAs expected to co-occur in the same subclone based on their pattern of evolution (chromosome 3, 4, and 9 CNAs in patient #1; chromosome 2 and 5 CNAs in patient #3; chromosome 7 and 11 CNAs in patient #4; Supplementary Fig. 30) tended to indeed co-occur in single cells based on scRNA-Seq inferred copy (Supplementary Figs. 15, 17–18), thus validating our scRNA-Seq inferred copy.

We also sought to validate these single-cell subclone assignments by determining whether subclone-specific SNVs had coverage in scRNA-Seq data. Though mutation detection was sparse, as seen by others[40], two subclone-specific SNVs in patient #3 had sufficient coverage in patient #3 scRNA-Seq data for analysis while other patients lacked adequate coverage (Supplementary Fig. 31; Methods section). Importantly, these two SNVs validated single-cell subclone assignments based on inferred scRNA-Seq CNA as they were only detected in cells predicted to harbor them based on CNA (Supplementary Fig. 17, bottom).

To identify how subclones present at the same timepoint differ in phenotype, we compared ssGSEA enrichment scores for 3331 C2 signatures, plus the all-58 RTKs signature[34] and our anti-apoptosis signature, between pre-treatment subclones by one-way ANOVA for each patient (Fig. 6a). We used pre-treatment (sensitive) timepoints for this analysis as the majority of post-treatment timepoints have a single subclone present following the bottleneck event. For reference, we also included the signature analysis on the post-treatment resistant subclone for these patients (Fig. 6b). The patient #1 pre-treatment survivor subclone that gave rise to the post-treatment bottleneck subclone had decreased antigen presentation and moderately increased EMT compared to the dominant dying subclone, suggesting that these phenotypes were at least partly genetic and pre-existed before treatment. B2M (β2-microglobulin) itself was

also statistically decreased in the pre-treatment survivor subclone compared to the dominant dying subclone (Supplementary Fig. 32). Thus, in patient #1, several pre-treatment subclones varied in antigen presentation and EMT levels (Fig. 6c). A subclone with decreased antigen presentation and partially increased EMT (light-orange) possessed a selective advantage and thus gave rise to the post-treatment bottleneck subclone (dark-orange), which had additionally evolved a progressively increased EMT state and increased RTK expression (Fig. 6c). Additionally, the patient #3 survivor subclone possessed an enhanced PI3K signature that was apparently carried into its descendant subclone after treatment (Fig. 6d). Finally, the patient #4 survivor subclone possessed decreased expression of transcriptional targets of the HOXA5 tumor suppressor[41], decreased TNF signaling, decreased antigen presentation, and increased RTK expression, which were likewise present in its descendant post-treatment subclones (Fig. 6e). These data indicate that in some cases the post-treatment EMT, RTK, and immune-avoidance phenotypes pre-existed before treatment in genetically defined subclones.

**Targeting evolving phenotypes using adaptive treatments.** It is important to identify adaptive therapeutic regimens that effectively target the dynamic, heterogeneous nature of breast cancer. Even where it is not possible to target a specific mutation, we can match treatments to altered phenotypes. Because RTK activation increased after bottleneck events in most of our patients (Fig. 5c, Supplementary Figs. 21–25), we hypothesized that post-treatment cells may have acquired sensitivity to drugs targeting this phenotype. To test this hypothesis, we compared the response of patient #1 pre- and post-doxorubicin cancer cells (days 305 and 732) to inhibitors of MEK and Akt (trametinib and MK2206[42]), two important RTK effectors[34]. Patient-derived breast cancer cells were grown using a modified fibroblast feeder approach, which has been shown to promote the growth of primary cancer

cultures, even from small tissue samples, compared to other methods[43, 44] (Methods section).

Post-doxorubicin patient #1 cells were moderately more sensitive to MEK and Akt inhibitors than pre-doxorubicin cells (Fig. 7a). We also tested these drugs in combination due to redundancy between PI3K and MAPK signaling[45]. Remarkably, post-doxorubicin cells were dramatically more sensitive to the equimolar combination of trametinib and MK2206 (Fig. 7b, top), with a combined IC50 of 4.37 μM in post-doxorubicin cells and unmet IC50 in pre-doxorubicin cells due to lack of response (see also Supplementary Fig. 33a). The combination was synergistic at 3.125 μM (Fig. 7b, bottom) and 6.25 μM (Supplementary Fig. 33b), in post-doxorubicin but not pre-doxorubicin cells. Further, as IC50-based drug response may be confounded by proliferation rates[46], we measured the effect of drug on proliferation rate in pre- and post-doxorubicin cells by comparing MEK + Akt inhibitor response in these cells at day 3 to the day 0 (day of drug addition) baseline (Supplementary Fig. 34). This revealed that, in post-doxorubicin cells, 3.13 μM combined equimolar dosing of trametinib and MK2206 was cytostatic, while the highest dose (6.25 μM) caused decrease in cell number over time, suggesting the possibility of clinical response to this combination. No dose caused change in proliferation rates of pre-doxorubicin cells, indicating acquired dependence on the MEK and Akt pathways only after treatment (Supplementary Fig. 34). These data suggest that enhanced RTK signaling post-doxorubicin (Fig. 5c) induced increased dependency on downstream PI3K and MAPK signaling. Therefore, when comparing drug efficacy between a patient's pre- and post-treatment timepoints, we find differential responses to drugs that correlate with the different phenotypes present in the pre-treatment and post-treatment timepoints. These findings indicate that the development of disease refractory to one treatment may be associated with enhanced sensitivity to an alternative treatment targeting post-treatment phenotypes. This approach can potentially be employed in drug-resistant cancers to improve outcomes.

## Discussion

Together, these data indicate that using transcriptional profiling to measure the broad phenotypic changes in patients can permit individualized therapeutic approaches to more effectively combat the dynamic and heterogeneous nature of cancer. While numerous genetic drivers likely promote acquired drug resistance, these genetic drivers likely converge on a smaller set of drug resistance phenotypes. Thus, phenotypic analysis can provide a feasible approach to more targeted care, even in situations where the precise genetic driver of cellular phenotypic change cannot be identified. Patient-specific tumor information could help the oncologist select therapy to better control the disease for longer periods of time and spare patients the toxicities from drugs that are unlikely to control disease due to emerging drug resistance.

The adaptive phenotype-targeted treatment approach we describe is analogous to the adaptive genotype-targeted approach used in lung cancer. Lung cancers with EGFR or ALK activating mutations generally acquire resistance to inhibitors targeting these proteins within 1 year due to secondary mutations that render the drugs ineffective or dysregulation of alternative RTK pathways[5, 47]. This has led to effective adaptive therapy and extended survival for many patients[48, 49]. These efforts were enabled through genetic analysis of post-treatment cancer specimens to determine the compendium of mutations promoting resistance. We propose that identification of the compendium of phenotypic changes occurring after treatment may likewise identify more effective treatment strategies. Further, targeting these phenotypic changes early in the disease progression, rather

than after outright resistance, may promote better outcomes, since even short-term drug treatment may induce phenotypes promoting drug resistance[50, 51].

Another strategy proposed for management of drug-resistant disease is maintenance of the index treatment, perhaps in combination with a new therapy or at intermittently high dosing, particularly in cases where partial sensitivity to the drug remains as with HER2 or EGFR inhibition[52, 53]. This strategy is likely to be most effective when the index treatment is relatively non-toxic, but may not be an effective long-term strategy for chemotherapy, where treatment duration and dosing can be limited by toxicity. In these cases, targeting the acquired phenotype alone without continued usage of the index therapy may be effective, as we have shown.

Our approach is designed to lead to clinically actionable strategies. Using multiple genetic and genomic tools to analyze heterogeneity of cancer in longitudinal samples delivers a mechanism to examine cancers as they change during treatment in actual patients. Current barriers to using this approach widely include the challenge of consistently re-sampling tumor samples during treatment and access to targeted agents currently not approved for breast cancer. However, we believe the technology described here can soon provide useful information to the oncologist that allows a 'proactive' approach to treatment choices instead of a 'reactive' approach. Utilizing individualized therapy along with additional drug regimens earlier in the treatment course may lead to prolonged stable disease and less drug resistance. In cases where drug resistance emerges, the oncologist could continue to customize treatments based on the molecular mechanisms currently driving tumor growth. Further, employing this strategy early in the treatment of breast cancer may prevent progression into metastatic disease.

By providing real time information specific to a patient's tumor, more informed decisions about continuing or modifying therapy can be made. Identifying common phenotypes of resistance to therapy will identify more generalizable targets for rational treatment of cancer patients. Ultimately, the genomic assessment of tumor subclones in real time combined with a dynamic approach of adaptive therapy that matches the tumor's capacity for evolution may benefit patient treatment strategies.

## Methods

**Statistical analysis**. scRNA-Seq ssGSEA enrichment scores and *B2M* expression were compared between pre- and post-treatment by Student's *t*-test (two-tailed). Within-sample subclone comparisons were done by one-way ANOVA. A two-proportion test was used to compare the number of scRNA-Seq cells expressing each gene between pre- and post-treatment (two-tailed). Fisher's exact test was performed on scDNA-Seq data to identify mutations co-occurring in single cells. *P*-value plots were generated by plotting $-1 \times \log_{10}$ (*P*-value), by two-tailed *t*-test or one-way ANOVA, on the *y*-axis against the ranks for these same values (higher ranks for lower *P*-values). Error bars for all drug assays represent standard deviation of four technical replicates from one assay. *P*-values were not adjusted for multiple comparisons; instead, analysis focused on lowest 5% of *P*-values when multiple comparisons were performed (Figs. 5b and 6a). Estimation of variance between groups, analysis of normality, and sample size estimation were not performed.

**Human subjects**. Informed consent was obtained from all patients in this study. Protocols were approved by the University of Utah Institutional Review Board.

**Samples and processing**. Breast cancer samples were obtained from malignant pleural effusions or ascites (or FFPE in one case). After fluid drainage, cells were pelleted at $3000 \times g$ for 5 min. Cells were resuspended in buffer (17 mM Tris, pH 7.4, and 135 mM $NH_4Cl$) and incubated at 37 °C for 5 min, followed by centrifugation, which was repeated until red blood cells were absent from pellet. Cells were washed thrice with PBS and frozen in 90% FBS with 10% dimethyl sulfoxide (DMSO). The patient #2 day 1320 pleural effusion had cancerous chunks from which DNA or RNA was isolated directly. For scRNA-Seq of this sample, minced chunks were dissociated in Renaissance medium (Cellaria) with 1% TrypLE (Life Technologies) and 1 U/μl DNase I and 2 mg/ml collagenase (Roche) for 20 min at

37 °C (shaking occasionally), followed by 10 min in a Biomaster 80 (Seward) and passing chunks through a 21-gauge syringe, and freezing in 10% DMSO for storage until scRNA-Seq.

**DNA/RNA isolation and sequencing.** Frozen vials of pleural effusion or ascites cells were thawed, and cells were pelleted to remove DMSO in supernatant and resuspended in buffer as described in Miltenyi OctoMACS and QuadroMACS instructions. CD45+ white blood cells were depleted from non-tumor-chunk samples (most samples) using the Miltenyi OctoMACS magnetic separation system with MS columns or the QuadroMACS magnetic system with LD columns along with anti-CD45 microbeads from Miltenyi. Germline DNA was obtained from peripheral blood mononuclear cells (PBMCs) from each patient or by sorting for CD45+ white blood cells from pleural effusions. DNA was isolated using Qiagen's QIAamp DNA Micro Kit, or for FFPE, the QIAamp DNA FFPE Tissue Kit. WES was performed at the Huntsman Cancer Institute's High Throughput Genomics Core Facility (HCI) using the Agilent SureSelect QXT Human All Exon v5 + UTRs kit and library prep and an Illumina HiSeq 2500 instrument with 125 cycles and paired-end sequencing. WGS was performed at the McDonnell Genome Institute at Washington University, NantOmics, or HCI using the Illumina TruSeq Nano DNA Library Prep Kit or PCR-free library prep and an Illumina HiSeq 2500 or X instrument with paired-end sequencing. RNA was isolated using Qiagen RNeasy Micro/Mini Kit and sequenced at HCI or NantOmics using Illumina TruSeq Stranded mRNA Sample Prep with oligo dT selection or TruSeq Stranded Total RNA Sample Prep Kit with RiboZero Gold library prep and Illumina HiSeq sequencing.

**Single-cell DNA sequencing.** Frozen viable pleural effusion or ascites vials were thawed and CD45+ white blood cells were depleted using Miltenyi's QuadroMACS. Individual cells were captured using Fluidigm C1 chips (10–17 µm cell diameter); whole-genome amplification was performed per manufacturer's instructions. Targeted amplification of mutation-containing regions was performed using Fluidigm Access Array and BioMark instruments per manufacturer's instructions. Libraries were sequenced on Illumina MiSeq. Reads were aligned to hg19 with BWA MEM v0.7.8. Variants were called using MuTect v1.1.4. Co-occurrence of mutations (at least 1 mutant read) in individual cells was evaluated using Fisher's exact test. Single cells with more than one subclone-defining mutation are shown.

**Whole-exome sequencing variant identification.** Read quality was verified using FastQC and reads were trimmed using Trimmomatic[54] v0.32 (http://www.usadellab.org/cms/index.php?page=trimmomatic). Alignment to hg19 was done with BWA MEM[55] v0.7.8 (https://github.com/lh3/bwa). BAM files were refined using PicardTools' *MarkDuplicates* and *FixMateInformation* tools, BamTools[56] *filter* tool (https://github.com/pezmaster31/bamtools), and GATK's[57] *RealignerTargetCreator*, *IndelRealigner*, and *BaseRecalibrator* (v3.2-2; https://www.broadinstitute.org/gatk/download/). WES somatic SNVs were called with MuTect v1.1.4 (https://www.broadinstitute.org/cancer/cga/mutect) and annotated with Oncotator v1.3.0.0 (https://github.com/broadinstitute/oncotator). CNAs were called from WES using VarScan[58] v2.3.7 (http://dkoboldt.github.io/varscan) copynumber and copyCaller, from pileups generated by SAMTools, to determine copy-number 2 genes for SubcloneSeeker. Segmentation was done with DNAcopy in R. Segmented data were converted to gene level using UCSC's refGene.txt (hg19) annotation. Log$_2$ fold-change values were converted to absolute copy using $2^{n+1}$ where n is log$_2$ fold-change from diploid. Copy number values were shifted in each sample to make the 2-copy peak centered at 2 and multiplied around the 2-axis to maximize number of genes near (within 0.1) of 1, 3, and 4 (to adjust for normal contamination).

**Whole-genome sequencing variant identification.** WGS DNA sequences were aligned to hg19 using BWA MEM using SpeedSeq. SNVs and indel variants were identified using FreeBayes[59] on each patient (see code repository for parameters). Variants were annotated using SnpEff and variants with quality below 5 were excluded. Somatic mutations were identified using "somatics" (https://github.com/brentp/gobio/tree/master/somatics), and only somatic mutations with germline VAF below 0.001 were used for subclone structure determination. As a secondary variant calling approach, we used VarScan somatic. This revealed the *APC* mutation in patient #3 not detected by FreeBayes; all other small variants discussed were identified with FreeBayes.

Structural variants were detected from WGS as follows. We used SAMBLASTER v0.1.22 to extract discordant paired-end reads and split reads. LUMPY[60] v0.2.12, or LUMPY within the SpeedSeq suite (*sv* utility), was used to call structural variants based on these reads, followed by SVTYPER (v0.0.2) to determine VAFs for each variant. Somatic variants were identified as with FreeBayes variants. Variants with quality <400 were excluded, except for patient #2 *ABCB1* fusions, which did not meet quality thresholds but were included due to corroborating identification in RNA-Seq data. Circos plots were made from FreeBayes, LUMPY, and CNA analysis. Circos plots show evolving mutations (going from VAF below 0.05 to 0.05 or above for small mutations and a threshold of 0.075 for structural) in color and truncal (ubiquitous) variants in gray; Cancer Gene Census[12] genes are shown by name, while other mutations are indicated by tick marks. Copy number alterations (CNAs) were determined from WGS data using VarScan as described for WES. For Circos plots CNA tracks, 30-segment window averages (absolute copy) were plotted after adjustment for normal contamination.

**Validation of variants in bulk RNA-Seq data.** Somatic SNVs identified by FreeBayes from WGS data were validated by detecting the SNVs in matched RNA-Seq data using UNCeqR[61, 62] in "interrogate" mode. RNA-Seq data were aligned using Rsubread as described in "RNA-Seq data processing." Each patient's somatic SNVs were input into UNCeqR in the form of a BED file encompassing the (trinucleotide) regions containing the mutations, along with the matched RNA-Seq BAM file for the same cancer sample. For each somatic SNV, UNCeqR determined the number of mutant and wild-type reads for each SNV, allowing determination of the RNA-Seq VAF. Only SNVs with germline VAF below 0.001, and in regions with at least 10-read coverage in RNA-Seq, were considered. 80.6–97.9% of each patient's somatic SNVs were detectable in RNA-Seq (Supplementary Fig. 1a), with excellent patient specificity (Supplementary Fig. 1b). Genomic fusion events identified by LUMPY[60] were corroborated by RNA-Seq.

**Validation of CNAs using SNP array.** One sample from patient #3 was analyzed for CNA by both WGS and SNP array for validation of WGS-based CNA calling pipeline (Supplementary Fig. 2). DNA was analyzed using the Illumina Infinium OmniExpressExome-8 v1.4 kit and the iScan system according to manufacturer instructions. Germline controls came from healthy individual data provided by the manufacturer. 30-SNP window averages (signal intensity) were used and regions containing measured SNPs were identified in WGS 30-segment window average data for comparison.

**Correction for normal contamination.** VAFs for WGS FreeBayes-identified somatic variants were corrected for normal contamination by determining tumor purity using CNA data. Absolute copy of each gene was determined using VarScan and DNAcopy in R as described in previous sections. Perfectly pure tumor samples have a large copy peak at 2 and smaller peaks at 1, 3, and 4, while normal-contaminated samples have a profile collapsing towards 2. We centered the largest peak at 2 and then multiplied values around the 2-axis using a range of multipliers until the maximum number of genes possible fell within 0.1 of 1, 3, and 4. The multiplier thus obtained was used to calculate tumor purity (=1.0/multiplier), and VAFs for FreeBayes somatic mutations were multiplied by the sample's multiplier to obtain adjusted VAF. VAFs used and reported are all adjusted except for structural variant VAFs, those in Supplementary Figs. 1 and 17 (bottom and legend), the germline *BRCA2* germline V643fs mutation VAFs in patient #4 given its presence in both normal and cancer cells, and patient #4 day 0 sample for which tumor purity was difficult to ascertain.

**Identification of evolution clusters and subclones from WGS.** Clusters of co-evolving mutations were identified from the WGS SNV and indel data (after normal contamination correction) by clustering copy-neutral (absolute copy 1.5–2.5 or, in the case of pseudo-tetraploid patient #2, between 3.6 and 4.4) mutations based on the samples in which they were present (at VAF of at least 0.05). Consensus VAFs for each mutation cluster were determined using kernel density estimation in R; CCFs for each cluster were then obtained by multiplying these values by 2 (or 4 for pseudo-tetraploid patient #2). In some cases two density peaks were present for a mutation cluster at a specific timepoint, indicating different sub-clusters and possibly additional subclones. Also, in some cases the CCF was determined based on the VAF of an index resistance mutation in the subclone, particularly when the CCF of the subclone was very low. Mutation clusters with the most mutations were incorporated into subclone analysis. Subclone structures were then determined using CCFs thus calculated, using rules described elsewhere[11]. For patient #1, an additional subclone was made apparent from deeper WES and scDNA-Seq (purple mutations in Fig. 2a) that was not detectable by WGS. Further, one patient #1 subclone (with light-orange mutations in Fig. 2a) had more subtle variation in CCF and its CCF was calculated based on the presence of a chromosome 9q amplification (i.e., average absolute copy of 2.5 in this region indicates 50% of cells with 3 copies and 50% of cells without the amplification). Subclone SC3's CCF in patient #2 was likewise estimated based on CNA.

**Identification of subclones using SubcloneSeeker.** We started with diploid, missense SNVs identified from deep WES by MuTect for patient #1 to corroborate WGS-based subclone findings. These variants were then subjected to Affinity Propagation Clustering[63] (via R package apcluster (https://cran.r-project.org/web/packages/apcluster/citation.html)), with similarity calculated via expSimMat($r = 2$, $w = 0.1$), to identify groups of variants that share similar VAFs across all samples. We then performed subclone structure reconstructions at adjacent pairwise timepoints through the pairwise joint analysis capability of SubcloneSeeker[11], and manually merged structures into a longitudinal evolution history, abiding to the same evolutionary consistency rule implemented in pairwise merging.

**Mutational signatures**. Mutational signatures were identified for WGS FreeBayes SNVs by first determining the trinucleotide sequence around SNV mutation sites, with the mutation site in the center, using an in-house script. For trinucleotides starting with an A or G, the reverse complement was used to minimize the number of mutation contexts as reported previously[16]. From these data we identified the percent of mutations that fell into each trinucleotide and transition/transversion pattern in each mutation cluster. These data were then used to determine COSMIC mutation signature weights using the deconstructSigs[17] package in R. COSMIC mutation signatures are defined at http://cancer.sanger.ac.uk/cosmic/signatures. Germline SNPs identified by FreeBayes were also analyzed for each patient using a similar approach.

**RNA-Seq data processing**. RNA-Seq data were processed with Rsubread[64] v1.16.1 (https://bioconductor.org/packages/release/bioc/html/Rsubread.html) in R using only uniquely mapped reads and the Hamming distance to break ties. The maximum indels allowed per alignment was 5. Gene-level expression values were processed to fragments per kilobase of transcript per million mapped reads (FPKM; bulk RNA-Seq) or transcripts per million (TPM; scRNA-Seq).

**Pathway predictions using ASSIGN**. EGFR, K-Ras G12V, and Akt pathway signatures were developed as described elsewhere[35] and can be found on GEO at GSE73628. RNA-Seq data was adjusted for batch effects using ComBat. Differentially expressed gene lists for each pathway were selected using ASSIGN[65] Bayesian gene selection. Gene lists were then used to estimate pathway activity in patient samples (see code repository for parameters). Molecular Signatures Database gene lists were also used as ASSIGN input in some cases.

**Single-cell RNA-Seq**. Frozen viable patient pleural effusions were thawed and CD45+ white blood cells, and in some cases fibroblasts (Anti-Fibroblast Microbeads, Miltenyi), were depleted using quadroMACS (Miltenyi). Cells were loaded into a Fluidigm C1 or C1 HT single-cell mRNA-seq chip (for 10–17 μm cell diameter), imaged via microscopy, and single-cell libraries were prepared per manufacturer's instructions using SMARTer chemistry. Illumina paired-end sequencing was performed and data were processed to TPM using Rsubread. Chambers containing more than one cell by microscopic imaging were excluded from analysis. Cells expressing fewer than 1700 genes or with fewer than 150,000 mapped reads were also excluded. CNAs were inferred from scRNA-Seq using the approach described elsewhere[24] using a window size of 101 genes. See code repository for complete details. Normal human mammary epithelial cells on which scRNA-Seq was performed using Fluidigm C1 were used as the normal (2-copy) state. Cells were assigned to subclones if they possessed CNAs associated with specific subclones. CNAs partially present pre-treatment, and increased post-treatment (i.e., increasing from absolute copy 2.2 to 3.0), were assumed to belong to the surviving subclone, while those partly present pre-treatment that disappeared were assumed to represent disappearing subclones. Cells were assigned to either survivor or disappearing subclones based on these CNAs. Validation of these subclone assignments was done by identifying subclonal (non-truncal) somatic WGS SNVs in scRNA-Seq data. UNCeqR was used for this purpose as with bulk RNA-Seq (see "Validation of variants in bulk RNA-Seq data"). SNV sites had to have at least 10 scRNA-Seq read coverage in a cell to make a call as to whether the cell was wild-type or mutant for the SNV, and SNVs with scRNA-Seq VAF never exceeding 0.2 in any cell were not analyzed. Two sub clone-defining 3′ UTR SNVs in patient #3 had coverage in at least one cell in each subclone and met the other criteria described above (in genes *BHLHE40*, defining surviving subclone SC6 and *DDX47*, defining surviving subclone SC7; median coverage of these SNVs was 215 and 91, respectively, in pre-treatment single cells with expression). Other patients lacked mutations meeting the above criteria. Violin plots of scRNA-Seq ssGSEA scores and TPM were generated with Seurat in R. As a secondary approach confirming our scRNA-Seq analysis using ssGSEA, we also performed PAGODA pathway over-dispersion analysis[28] on scRNA-Seq cells from each patient (cancer cells only) using R, with transcript count data as input rather than TPM.

**Heatmaps**. RTK heatmaps were generated by adding the minimum non-zero FPKM expression value for the RTKs to FPKM values for every RTK and calculating $\log_2$ fold change from day 0. Heatmap was generated using Complex-Heatmap in R.

**ssGSEA**. ssGSEA was run using GSVA v1.14.1 in R (https://www.bioconductor.org/packages/release/bioc/html/GSVA.html) using RNA-Seq FPKM (bulk) or TPM (single-cell) values and Molecular Signatures Database C2 (version 5) signatures and custom signatures described. Full EMT signature names described in the text are ANASTASSIOU_CANCER_MESENCHYMAL_TRANSITION_SIGNATURE, JECHLINGER_EPITHELIAL_TO_MESENCHYMAL_TRANSITION_UP, and HALLMARK_EPITHELIAL_MESENCHYMAL_TRANSITION; immune signatures in Fig. 5c are PHONG_TNF_TARGETS_UP and REACTOME_ANTI-GEN_PRESENTATION_FOLDING_ASSEMBLY_AND_PEPTIDE_LOADING_OF_CLASS_I_MHC; proliferation signatures are SA_REG_CASCADE_OF_CYCLIN_EXPR, KALMA_E2F1_TARGETS, and KONG_E2F3_TARGETS. Our

custom anti-apoptosis gene set consisted of the 6 anti-apoptotic Bcl-2 family members[66] *BCL2, BCL2L1, BCL2L2, MCL1, BCL2L10* and *BCL2A1*, and the 8 inhibitors of apoptosis family members[67] *NAIP, BIRC2, BIRC3, XIAP, BIRC5, BIRC6, BIRC7*, and *BIRC8*. The all-58 RTKs signature was assembled from a review[34].

**TopHat fusion**. TopHat[68] v2.0.6 (http://ccb.jhu.edu/software/tophat/index.shtml) was run on bulk RNA-Seq data using hg19 followed by TopHat-Fusion Post (see code repository for parameters). This identified the *SLC25A40-ABCB1* fusion transcript in patient #1 but missed the *ABCB1* fusions in patient #2, which we identified manually by searching for reads fusing the 3′ end of *ABCB1* exon 2 with transcripts aligning elsewhere (Blast search).

**Western blots**. Western blots were performed by scraping cells in cold PBS and lysing pelleted cells in lysis buffer (5 mM EDTA, 150 mM NaCl, 50 mM Tris, pH 8.0, 1% Triton X-100, and 0.1% SDS with protease and phosphatase inhibitor cocktails from Sigma) for 15 min on ice. Cleared diluted supernatants were boiled in Laemmli buffer 10 min and 20–30 μg of protein per lane was run by SDS-PAGE and transferred to polyvinylidene difluoride membrane. Western blot on untreated patient #1 cells was done on never-passaged cells and assay was performed once. Cells were cultured in Renaissance medium. Antibodies used were from Cell Signaling and the catalog numbers were: #5741 (vimentin), #4695 (Erk1/2), #9272 (Akt), #4267 (EGFR), #4370 (p-Erk1/2), #4060 (p-Akt), and #2234 or #3777 (p-EGFR).

**Immunofluorescence staining**. Patient #1 never-cultured pleural effusion or ascites cells were plated at ~3,000 cells/well in 384-well black plates with clear bottom in Renaissance medium (Cellaria) with 5% FBS, 25 ng/ml cholera toxin (Sigma), and 1% antibiotic/antimycotic (Life Technologies); to this was added 20% filtered pleural effusion fluid from another patient. After 3–4 days of culture, cells were washed twice with PBS and fixed in 2% paraformaldehyde (Electron Microscope Sciences) in PBS for 15 min. Cells were washed thrice with PBS and permeabilized in 0.1% Triton X-100 in PBS. Permeabilized cells were incubated with primary antibody in 2% bovine serum albumin overnight at 4 °C. Samples were washed twice with PBS containing 0.05% Tween-20, then incubated with Alexa-conjugated secondary antibodies (Thermo Fisher) and DAPI (Invitrogen) for one hour. Samples were washed twice and stored in PBS at 4 °C. Imaging was performed using an automated high-throughput fluorescence microscope (Olympus scanR). This experiment was performed one time.

**DNA content measurement**. Frozen viable pleural effusions were thawed and washed. Cells were resuspended in 4% paraformaldehyde in PBS and fixed 15 min. Cells were pelleted and resuspended in 5 ml PBS for 15 min. Cells were pelleted and resuspended in staining buffer (2% FBS in PBS). 3 μM DAPI was added and incubated 15 min. Cells were washed twice in PBS and resuspended in PBS, followed by flow cytometry.

**Cell culture and drug assays**. The fibroblast feeder system used for drug assays was modified from that reported previously[43]. We plated 800 irradiated mouse embryonic fibroblasts and 5600 never-cultured patient-derived cells ("cancer + fibroblast" assay), thawed from frozen viable state, per well in white 384-well plates (quadruplicate). Control wells with 800 fibroblasts per well and no cancer cells were also plated ("fibroblast-only" assay). Medium included 10 μM Y-27632[43]. 2–3 days later drug was given. 3 days after drug addition, viability was measured using CellTiter-Glo. Cancer + fibroblast signal and associated fibroblast-only signal are shown in Supplementary Fig. 33a, while cancer + fibroblast minus fibroblast (to obtain cancer-only signal) are shown in Fig. 7a, b and Supplementary Fig. 34. Data were normalized to dimethyl sulfoxide treatment mean for cancer + fibroblasts or cancer + fibroblast minus fibroblast. IC50's for cancer + fibroblast minus fibroblast were generated in GraphPad Prism using regression analysis. Synergy was calculated by Bliss independence (Fig. 7b, and Supplementary Fig. 33b). A two-sided *t*-test was performed to compare actual combination response with expected response. Patient Western blot (Supplementary Figs. 24b, 27) was done in Renaissance medium and was performed one time.

**Code availability**. Custom code used can be found at https://bitbucket.org/samuelwb/tumorhetcode.

**Data availability**. Somatic mutation data and scRNA-Seq gene expression data for each patient are available on the European Genome-phenome Archive (EGA) under accession EGAS00001002436, with controlled access, the details of which can be found on the website under this accession. Custom gene expression signatures can be found on GEO at GSE73628.

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

## Acknowledgements

We thank the anonymous patients used in our study for their generous contributions to research. We acknowledge Patricia Bild for inspiring this research. We also thank the Huntsman Cancer Institute Biorepository, which collected most of the biospecimens in this study. We acknowledge Dr. David Bowtell for helpful feedback on this project. S.W. B. was supported by NLM training grant 5T15LM007124. This research was supported by NIH 4U01CA164720-05 and U54 CA 209978-01 to A.H.B. and U24CA209999 to G.T. M. We also acknowledge University of Utah core facility experimental support, including the flow cytometry core (funded by NIH grant 1S10RR026802-01) and cell imaging core (funded by NIH grant 1S10RR024761-01).

## Author contributions

S.W.B. contributed study design and coordination, wrote manuscript, data processing, and analysis including WGS, subclone identification, and RNA-Seq analysis, biochemical assays, drug assays. J.A.M. performed molecular studies, drug assays, biochemical assays, and contributed to manuscript writing. Y.Q. contributed subclone structure determination, data analysis, and contributed to manuscript writing. S.R.P. performed RNA-sequencing analysis and processing exome sequencing data, design of study. G.S. performed processing of DNA samples, drug assays, western blots, and tumor dissociation. R.M.L. contributed structural variation analysis from WGS data. B.S.P. contributed single-nucleotide and insertion–deletion analysis from WGS and WES data. A.E. performed immunofluorescence assays. J.S.P., S.R.S., and R.H.M. performed RNA-Seq analysis. L.A.A. contributed clinical support and infrastructure. B.K.D. ran scRNA-Seq experiments. R.E.F. performed pathological analysis of cancer specimens. C.B.R. contributed clinical oversight and provided essential reagents. J.P.B. contributed clinical oversight and provided essential reagents. D.Y.L. contributed study coordination. P.J.M. contributed study design. L.M.H. and J.W.G. contributed study design and immunofluorescence assays. W.E.J. performed RNA-Seq analysis, study design and coordination, and contributed to manuscript writing. D.F.J. contributed RNA-Seq analysis. S.S.B. contributed clinical oversight, provided essential reagents, and contributed to manuscript writing. A.L.C. contributed clinical oversight, provided essential reagents, and contributed to manuscript writing. A.R.Q. contributed WGS data processing and analysis, CNA analysis, study design and coordination, and contributed to manuscript writing. G.M. contributed WGS data processing and analysis, mutation, study design, and coordination. T.L.W. conceived the project, coordinated the study, and contributed to manuscript writing. A.H.B. conceived the study, contributed study design and coordination, wrote manuscript, and performed data analysis.

## Additional information

**Competing interests:** The authors declare no competing financial interests.

**Change history:** A correction to this article has been published and is linked from the HTML version of this paper.

