## [Peer Review File · Nature Communications]

Reviewers' comments:

Reviewer #1 Expert in prostate cancer transcriptomics:

The paper by Brady et al performs subclonal reconstruction of 4 breast cancers, followed longitudinally, and profiled through a combination of whole-genome sequencing, exome sequencing and (targeted) single cell sequencing. The authors subsequently perform single-cell RNA sequencing and match the cells to normal cells and tumour cells (including different subclones) through copy number profiling, and then assay phenotypes of treatment resistance, such as EMT, RTK expression, antigen presentation. They also culture the cells, expose them to treatment and correlate cell phenotypes with treatment susceptibility. In a way the study represent a series of deeply worked out case reports, in which not only the tumour evolution is studied used genomics, but the authors also link this with cell phenotypes and treatment. As such the study aims to illustrate how personalised genomics can be done in the future. I found the study quite impressive - it is a genuine tour de force. The slight downside is that as presently written, the study is not very accessible to the reader, and while the paper is quite wordy, it is not always clear what exactly was done, how well it worked, and how causal some of the links the authors make are, as detailed below.

Main comments:

1. It is not entirely clear to me which samples were whole-genome sequenced and exome sequenced, and particularly how the targeted single-cell DNA sequencing experiment was designed. Is it correct that whole-genome sequencing was performed on 4 samples for patient one, 6 from patient 2, 2 from patient 3 and 4 samples from patient 4 (+ each time germline), and exome sequencing was only performed for patient 1 as a validation? In the targeted single-cell DNA sequencing, I'm assuming the mutations called from the whole-genome sequencing experiment were targeted and genotyped. How many were targeted? Where does Mutect come in here? Is it correct no new mutations were called on the single cells (i.e. mutations were genotyped, not called de novo)?
2. What I'm missing is validation of the author's pipelines. What's the sensitivity/specificity of mutation calling? Interestingly, the single-cell data could be used to validate called mutations.
3. Several of the links of genotypes/phenotypes to treatment the authors make are hypotheses, and the causal link is unclear. E.g. the increase of APOBEC associated mutations in patient #1 SC1 linked to increased APOBEC3B expression after paclitaxel treatment. The mutations are increased and APOBEC3B shows somewhat higher expression after cells are treated with paclitaxel - but any link between both is purely a hypothesis.
4. It would be worthwhile to deconvolve mutational spectra (which is what the authors really show here) into activities of known mutational signatures. This would actually also be an indirect validation of the author's pipelines (including mutation calling).

Minor comments:

5. Is the DNA sequencing data available through EGA (it should be made available) - only the RNA seq data is mentioned in that regard (GSE73628).
6. How were the patients selected? Three are BRCA2-positive, which is not that common.
7. Similarly, two were HER2+. Do the authors find the HER2 amplification? Is it present in all subclones?
8. I would avoid using the term CNV for somatic copy number alterations, as CNVs strictly refer to copy number variants, present in normal tissue and polymorphic in a population. I'd go for CNA or sCNA.
9. The term 'background' mutations is confusing. I'd go for 'truncal mutations' (and separate out those that are in the germline)
10. The authors use the term cellular prevalence to describe fractions of tumour cells in a certain subclone. Cellular prevalence refers to the fraction of cells in a population compared to the total number of cells. Cancer Cell Fraction (CCF) would be the correct term here, as that refers to the

fraction of *cancer* cells (rather than all cells).

11. While the approach to map the single cell RNA seq data to tumour/normal cells and to subclones is very neat and from the supplementary figures seems to work by and large, it is unvalidated. It can be validated by leave-one-out analysis for example, leaving out one region in the classification and using that to see if the results are consistent.

12. What exactly is shown in Supplementary Fig. 6? Is this from bulk sequencing, or is through e.g. FACS? Could the small peak annotated as 1* in Supp. Fig 6b actually correspond to 0 copies (due to normal contamination)?

13. It would be worthwhile redrafting the abstract, to make it more accessible to a general audience.

Reviewer #2 Expert in cancer evolution:

In this article, the authors present data and analysis for 4 breast cancer patients that were observed for multiple years each. For each patient, the authors collected samples at several time points to follow the evolution of these tumors over time. They performed DNA and RNA sequencing as well as single cell RNAseq on these samples. They identified several mutations and alterations that were accumulated over time and identified partly known and partly novel resistance mechanisms. They also performed cell line experiments to investigate whether different treatments might improve therapeutic outcomes.

Overall I thought this was a fantastically interesting read, one of the more interesting papers in breast cancer evolution lately. Particularly the ability to analyze several samples from the same patient and over many years makes this an interesting contribution. Some findings will be very important for the field, for instance the fact that BRCA2 mutations might be reversible during the course of tumor evolution.

I do have a few issues, mostly technical:

- there are many ways to perform RNAseq, and the Fluidigm chip the authors used is just one of them; the article lacks a discussion of comparison across methods, why they chose the Fluidigm chip, and whether and how the use of this methodology might bias the findings. the Teichmann group recently published a Nature Methods paper comparing platforms, which might be helpful.

- it is well known that single cell RNAseq has high dropout rates (up to 80%). how did the authors deal with this issue?

- similarly, batch effects in single cell RNAseq are large and introduce artifacts that can render analyses completely misleading. this issue is not mentioned and apparently has not been addressed when analyzing the samples

- the CNA calling algorithm the authors used to analyze their single cell RNAseq data (from Aviv Regev's lab, the melanoma science paper) is very sensitive to the choice of window sizes. The current paper neither describes which window size was used (Im guessing the default, 100), nor were sensitivity analyses performed to study the effects of this choice. It is a randomly chosen parameter that very significantly determines the results so needs to be handled with care.

- related to the above comment, there might be better ways to identify tumor cells apart from using a quite crude CNA caller together with clustering. For instance, specific driver mutations could be used.

- the authors state that they identify point mutations in single cell RNAseq data by aligning with mismatches and keeping a mutation if found in two cells. these are also arbitrary choices that will

determine outcomes. please carefully study the effects of these.

- the discussion of adaptive therapy needs to be handled with more care. the authors cite the work by gatenby et al but this is a controversial idea, and also not the only method that identifies treatments based on systems approaches (see methods by chmielecki et al, Science Translational Medicine 2011 and others). please present an inclusive discussion of approaches

Reviewer #3 Expert in systems biology of cancer:

Bild and collaborators describe an experimental tour de force, in which they follow the clonal evolution of metastatic ER+ breast cancer (BrCA) tumors in 4 patients over several years (respectively, 2, 3, 3 and 15 years follow-up), in response to various sequential treatments. By a combination of bulk and single-cell RNA-Seq, they were able to collect homogeneous datasets that were then analyzed with insightful and appropriate software packages (e.g., SubcloneSeeker).

The overall strategy is straightforward and well explained in Fig.1, but logistic and technical challenges should not be underestimated. They were resolutely overcome, producing a reliable dataset that, in my opinion, provides a roadmap to future studies and may well become a classic model for precision oncology approaches based on patients' real time data.

There are several important conclusions emerging from this study, which will be of interest not just to the clinical oncologist, but also to cancer researchers at large. Moreover, cancer systems biologists will find this dataset quite intriguing, since it begins to unravel real-life strategies of clonal competitive survival and selection trajectories.

The most informative patient maybe #4, due to transition from DCIS to metastatic disease. Here, the role of genetic mutations (SNVs or indels) is strongly supported by the data.

Other key findings include bottleneck events, such as the ABCB1 promoter fusions which may well be a common event in resistance to doxorubicin, and mutant BRCA2 reversion events, which I find unexpected and fascinating. Extensive data analyses lead the authors to propose "mutational signatures" as a metric for clonal identification, possibly supporting the conclusion that treatment selects for clones with a signature that suggests resilience to immune system attack. This will be a topical finding for the currently hot field of immune therapy.

A few points should be addressed:

1. The relationship between clonal mutational signatures and clonal fitness could be discussed in more depth, with reference to existing literature on the subject of cancer evolutionary dynamics, which is extensive. In particular, it is not clear what is meant for "clonal fitness capacity" (line 210).
2. Cancer clone phenotypes are evaluated functionally in vitro (lines 319 and on), in terms of their behavior in response to various drug. However, in these assays the potency of the drug is measured, not the proliferative capacity of cells in the presence of those drugs. The authors should discuss this point with reference to recent publications highlighting instead the use of proliferation rate metrics ([1] Nat Methods. 13 (2016) 521-527, and 497-500). Such metrics would be quite appropriate for attempts to predict dynamics of clonal expansion, and possibly time to relapse.
3. A minor technical point: the role and/or necessity for feeder cells in the drug assays should be explained.

Note: We have numbered and highlighted in yellow each comment in the form R1-1, R1-2 to indicate reviewer #1 point #1, reviewer #1 point #2, and so on. Some reviewer comments are subdivided using letters. After each reviewer comment, we also indicate the locations of changes in the manuscript, followed by the response.

Reviewer #1 Comments

R1-1a: “It is not entirely clear to me which samples were whole-genome sequenced and exome sequenced, and particularly how the targeted single-cell DNA sequencing experiment was designed. Is it correct that whole-genome sequencing was performed on 4 samples for patient one, 6 from patient 2, 2 from patient 3 and 4 samples from patient 4 (+ each time germline), and exome sequencing was only performed for patient 1 as a validation?”

Locations of manuscript changes: Fig. 2 and lines 74-75, 82-84.

Yes, the reviewer correctly understands which samples were sequenced by which methods. To clarify which samples were analyzed by which approaches, we have added symbols to Fig. 2 next to each sample to indicate the analyses performed on the sample.

The targeted single-cell DNA sequencing experiment is also explained in more detail in the revised manuscript text. Briefly, 17 mutations identified from exome sequencing of patient #1 were analyzed, with a focus on subclonal mutations to resolve subclone structure. Single-cell DNA was captured via Fluidigm C1, regions surrounding the mutations were amplified using custom primers, and next-generation sequencing was performed to detect whether the mutations were present in each single cell.

R1-1b: “In the targeted single-cell DNA sequencing, I’m assuming the mutations called from the whole-genome sequencing experiment were targeted and genotyped. How many were targeted? Where does MuTect come in here? Is it correct no new mutations were called on the single cells (i.e. mutations were genotyped, not called *de novo*)?”

Locations of manuscript changes: Lines 82-84.

Yes, in the targeted single-cell DNA sequencing experiment, mutations detected using bulk DNA sequencing (exome sequencing in this case) were targeted and analyzed. Our exome sequencing data had greater depth (100×) than whole-genome sequencing (60×) and thus was able to detect rarer subclones. Thus, some of the mutations selected for single-cell DNA sequencing were only present in the exome sequencing analysis due to lower allele frequency.

As mentioned above, 17 mutations were targeted in the single-cell DNA sequencing experiment. MuTect was used to determine the allele frequency of the 17 mutations in each cell after next-generation sequencing of the PCR-amplified regions surrounding each mutation.

It is correct that no new mutations were called on the single cells. Only the specific regions surrounding the 17 mutations were analyzed. We have clarified this in the revised manuscript text.

R1-2: “What I’m missing is validation of the author’s pipelines. What’s the sensitivity/specificity of mutation calling? Interestingly, the single-cell data could be used to validate called mutations.”

Locations of manuscript changes: Supplementary Figs. 1, 2, 8, 12; lines 60-69, 127-129, 200-202.

In our revised manuscript, we have validated our mutation calling pipelines using the following methods:

- *RNA-Seq.* We detected somatic SNVs identified from whole-genome sequencing (WGS) in matched bulk RNA-Seq from the same sample using UNCeQr, which showed that 80.6% to 97.9% of WGS SNVs in expressed regions could be detected in RNA-Seq, including driver mutations (new Supplementary Fig. 1).
- *SNP array.* We analyzed one patient’s copy number alterations by SNP array and found this secondary approach to be highly concordant with our copy number analysis inferred from WGS (new Supplementary Fig. 2).
- *Mutation signature analysis.* We performed mutation signature analysis on each patient’s somatic SNVs and germline SNPs (all identified from WGS) and found that only somatic SNVs had cancer-associated BRCA and APOBEC signatures, while the germline SNPs lacked these signatures. Further, patients with *BRCA2* mutations had higher BRCA mutation signatures (new Fig. 4b and Supplementary Fig. 8).
- *Comparison with clinical testing.* Our identification of *BRCA2* germline variants in 3 of 4 patients matched clinical *BRCA1/BRCA2* testing, which was performed in all 4 patients.
- *Direct visualization of reads.* We show from patient #4 sequencing reads that the patient’s *BRCA2* germline variant and somatic reversion are in *cis*, thus leading to a restoration of frame, as they are present on the same reads (new Supplementary Fig. 12).
- *Side-note validations.* We also note in our revised manuscript that specific genomic fusions events identified by WGS were also found within RNA-Seq data. For example, fusions in intron 1 of the *ABCB1* gene, identified by WGS, are corroborated by RNA-Seq fusions reads from nearby exons brought near one another by the genomic fusion event (Supplementary Figs. 6, 11, which were included in the original manuscript).

This information is described in a new Results section titled “Validation of variants identified.”

R1-3: “Several of the links of genotypes/phenotypes to treatment the authors make are hypotheses, and the causal link is unclear. E.g. the increase of APOBEC associated mutations in patient #1 *SC1* linked to increased *APOBEC3B* expression after paclitaxel treatment. The mutations are increased and *APOBEC3B* shows somewhat higher expression after cells are treated with paclitaxel – but any link between both is purely a hypothesis.”

Locations of manuscript changes: Deletion of text and of one Supplementary Figure.

We agree and have accordingly removed these results and the associated Supplementary Figure, as deep mechanistic studies on how the APOBEC mutation signature was enriched in certain subclones are outside the scope of our study.

R1-4: “It would be worthwhile to deconvolve mutational spectra (which is what the authors really show here) into activities of known mutational signatures. This would actually also be an indirect validation of the author’s pipelines (including mutation calling).”

Locations of manuscript changes: Fig. 4b, Supplementary Fig. 8; lines 117-131, 158-161, 174-177, 224-229.

We thank the reviewer for this helpful comment, which led us to implement deeper mutation signature analysis (in addition to mutation context/spectra analysis, which was included in our original manuscript) for each subclone using the *deconstructSigs* package in R (new Fig. 4b and new Supplementary Fig. 8).

This broadened our findings in relation to mutational processes, confirmed the presence of the APOBEC mutation signature in some subclones, and revealed BRCA-deficiency-associated (homologous recombination deficiency) mutational processes in our 3 *BRCA2*-deficient patients, thus indirectly validating our mutation calling pipelines as suggested by the reviewer. Further, germline SNPs identified from WGS lacked the APOBEC and BRCA-deficiency mutation signatures; these signatures were only detected in WGS somatic SNVs (Supplementary Fig. 8). This serves as an additional validation of our mutation calling pipelines, as germline variants would not be expected to harbor these cancer-associated mutational processes.

R1-5: “Is the DNA sequencing data available through EGA (it should be made available) – only the RNA seq data is mentioned in that regard (GSE73628).”

Locations of manuscript changepubs: Lines 746-750.

We have accordingly deposited the DNA sequencing and single-cell RNA sequencing data on EGA under accession EGAS00001002436. This accession is noted in our new Methods section titled “Data availability.”

R1-6: “How were the patients selected? Three are *BRCA2*-positive, which is not that common.”

Locations of manuscript changes: Lines 52-53, 66-68.

This is an insightful comment by the reviewer, as it is expected that ~12% of unselected patients will have *BRCA1/2* mutations. Patients were included in our study based on the availability of repeated cancer samples over time in the form of cancer-rich metastatic pleural effusion and ascites fluids. Pleural effusions occur in some 10-20% of breast cancer patients, and the presence of repeated pleural effusions over years of time is relatively rare. As the reviewer indicates, we have begun to ask on a broader scale if *BRCA2*-deficient cancers have an increased tendency to metastasize to the pleura in a recurrent manner. Note that *BRCA2* mutation status was determined both by clinical testing and confirmed by our WGS, indicating that the *BRCA2* variants identified were not artifactual.

In our revised manuscript we have included a statement explaining the selection of patients, as described above, in the Results section titled “Patient treatment history and approach.” We have also noted in our new section titled “Validation of variants identified” that our WGS matched clinical *BRCA2* testing performed in our patients.

R1-7: “Similarly, two were *HER2+*. Do the authors find the *HER2* amplification? Is it present in all subclones?”

Locations of manuscript changes: Supplementary Fig. 5 and lines 101-102, 142.

Yes, we detected moderate *ERBB2* (*HER2*) amplification in patient #1 (3 copies) and patient #2 (5 copies; see new Supplementary Fig. 5), who were determined to be clinically *HER2+*. *ERBB2* amplification was present in all subclones of these patients and chromosome 17q amplification (where *ERBB2* resides) could be observed in single-cell RNA sequencing (inferred copy) data in every cancer cell from these two patients (Supp. Figs. 14-15) but not in our *HER2-* patients (Supp. Figs. 16-17).

Patient #1 received trastuzumab and the experimental *HER2/HER3* antagonist MM-111, as indicated in Fig. 2a, though response was minimal. The clinical *HER2* testing on this patient was done early in the patient’s disease history, and the copy number may have been higher than 3 at earlier stages in the patient’s treatment.

R1-8: “I would avoid using the term CNV for somatic copy number alterations, as CNVs strictly refer to copy number variants, present in normal tissue and polymorphic in a population. I’d go for CNA or sCNA.”

Locations of manuscript changes: Throughout.

We thank the reviewer for the correction. We have accordingly replaced “CNV” with “CNA” in our revised manuscript text and figures.

R1-9: “The term ‘background’ mutations is confusing. I’d go for ‘truncal mutations’ (and separate out those that are in the germline).”

Locations of manuscript changes: Throughout.

We have accordingly replaced the term “background mutations” with “truncal mutations” in our revised manuscript text and figures.

Also, we have now considered germline *BRCA2* variants as non-truncal (since they are germline) in our revised manuscript as suggested, which we agree will clarify the manuscript. In Fig. 2 we refer to germline *BRCA2* mutations as “g*BRCA2*” to separate out germline from somatic *BRCA2* variants. We do, however, refer to loss-of-heterozygosity of these germline variants as truncal events (since the loss-of-heterozygosity is a somatic event). Thus “g*BRCA2* (E49*)” refers to a germline heterozygous variant in Fig. 2b, while “*BRCA2* (E49*) LOH” refers to a truncal somatic event.

R1-10: “The authors use the term cellular prevalence to describe fractions of tumour cells in a certain subclone. Cellular prevalence refers to the fraction of cells in a population compared to the total number of cells. Cancer Cell Fraction (CCF) would be the correct term here, as that refers to the fraction of *cancer* cells (rather than all cells).”

Locations of manuscript changes: Throughout.

We thank the reviewer for the correction. We have accordingly replaced the use of the term “cellular prevalence” with “Cancer Cell Fraction” or “CCF” throughout our revised manuscript text and figures.

R1-11: “While the approach to map the single cell RNA seq data to tumour/normal cells and to subclones is very neat and from the supplementary figures seems to work by and large, it is unvalidated. It can be validated by leave-one-out analysis for example, leaving out one region in the classification and using that to see if the results are consistent.”

Locations of manuscript changes: Fig. 6, Supplementary Figs. 14-17 (updated) and 18, 29 (new);
lines 240-241, 307-326, 335-349

Thank you for your insightful comment. In our revised manuscript we have validated our assignment of single cells to subclones using single-cell RNA sequencing (scRNA-Seq) inferred copy number, and increased the stringency of subclone assignment of single cells, as follows.

- *Gene expression profiles of normal vs. cancer.* In our revised manuscript we show data indicating that cells inferred to be normal by scRNA-Seq copy number analysis (non-copy-altered) express much higher levels of normal cell markers than cells inferred to be cancer (copy-altered). For example, mesothelial cells and fibroblasts are common normal cells present in pleural effusions (the sample type on which we performed scRNA-Seq) and we find that the inferred-normal cells express dramatically higher levels of the mesothelial cell marker *PDPN* (podoplanin) than the inferred-cancer cells (new Supplementary Fig. 18).
- *Corroboration with point mutation data.* In our revised manuscript we have performed a comprehensive analysis of subclone-specific SNVs in our scRNA-Seq data for each patient, which previously was done only on patient #1 semi-manually on a subset of mutations, using the UNCeQR tool. Though coverage of these SNVs was extremely poor in scRNA-Seq data, largely due to the 3'-biased method of library preparation used for scRNA-Seq (poly(A)-based), two subclone-specific 3' UTR variants had high coverage in patient #3 scRNA-Seq data. Further, these were the only mutations with coverage in at least one cell from all subclones (as assigned by scRNA-Seq copy profile). These two mutations were only detectable in single cells expected to harbor them based on

scRNA-Seq inferred copy number (updated Supplementary Fig. 16, bottom), thus partly validating our approach to subclone assignment.

- *Corroboration using multiple CNAs.* In our revised manuscript we have comprehensively identified all subclonal CNAs from WGS that could be used to aid in subclone identification in scRNA-Seq data (new Supplementary Fig. 29). This led to identification of two additional subclonal CNAs in patient #1 aiding in subclonal identification, which refined our subclone assignments and enabled us to assign some previously unassigned cells to specific subclones (updated Supplementary Fig. 14 and Fig. 6a-c). (Incidentally, we have also removed cells of uncertain subclone assignment, previously labeled “?” from our pathway signature analysis in our original Fig. 6b, to improve clarity.) We have also increased the stringency for assignment of subclones to cells, requiring multiple patient CNAs to assign cells to a subclone, which led to exclusion of patient #2 from this analysis since subclones were assigned based on a single CNA (chromosome 20; see updated Fig. 6 and Supplementary Fig. 15). These results have not changed our fundamental conclusions, as we find decreased antigen presentation and TNF signaling and increased EMT and RTK signaling in pre-treatment survivor subclones. The presence of multiple CNAs expected to co-exist in a single cell, and not in others, offers stronger corroboration of single-cell assignment as in a few cases cells were assigned to a subclone based on a single CNA. Our comprehensive identification of subclonal CNAs from WGS has also revealed the important validations that truncal CNAs (by WGS) tend to be present in all single cells by scRNA-Seq copy number analysis, while subclonal CNAs (by WGS) tend to only be present in certain populations of single cells and co-occur in scRNA-Seq single cells (inferred copy) with other CNAs expected to be in the same subclone by WGS, which we have noted in our revised manuscript.

R1-12: “What exactly is shown in Supplementary Fig. 6? Is this from bulk sequencing, or is through e.g. FACS? Could the small peak annotated as 1* in Supp. Fig 6b actually correspond to 0 copies (due to normal contamination)?”

Locations of manuscript changes: Lines 137-138 and Supplementary Fig. 9 legend.

We note that our original Supplementary Fig. 6 is now Supplementary Fig. 9 in the revised manuscript, and refer to it by its new numbering.

Supplementary Fig. 9a-c were obtained from bulk DNA sequencing, and Supplementary Fig. 9d shows flow cytometric quantification of DNA content. We have clarified this in the revised manuscript text describing this figure and in the figure legend.

The small peak annotated 1* (with the asterisk referring to inferred absolute copy) in Supplementary Fig. 9b could not “correspond to 0 copies (due to normal contamination)”, since these plots show absolute copy rather than \log_2 fold change from diploid. Therefore, normal contamination would push the peaks towards absolute copy 2* (or 4* if pseudo-tetraploid). For example, in a 50% cancer, 50% normal sample the peaks for 1* and 3* would be at 1.5 and 2.5, respectively, due to the presence of diploid (2-copy) normal cell contaminants. That is, all copy-altered regions would move closer to 2 as a result of the normal cell contaminants, not closer to 0, because of the averaging effect of having cells with 2 copies at every locus.

As a tertiary verification of patient #2’s pseudo-tetraploid status, note that in the Fig. 2b heatmap showing this patient’s variant allele frequencies, at the bottom are a group of truncal mutations that are dark-blue (variant allele frequency around 0.5, or 2/4 copies mutated), representing mutations occurring before the genome doubling occurred (~50% of truncal somatic variants). Above this are lighter-blue truncal variants that likely occurred after the genome doubling (variant allele frequency around 0.25, or 1/4 copies mutated), representing the other ~50% of truncal somatic variants). This pattern is not observed in the other patients and is consistent with pseudo-tetraploid status determined using other means.

R1-13: “It would be worthwhile redrafting the abstract, to make it more accessible to a general audience.”

Locations of manuscript changes: Lines 2-14.

We have accordingly revised the abstract to provide additional background and perspective, remove unnecessary jargon, and clarify meaning.

Reviewer #2 Comments

R2-1: “There are many ways to perform RNAseq, and the Fluidigm chip the authors used is just one of them; the article lacks a discussion of comparison across methods, why they chose the Fluidigm chip, and whether and how the use of this methodology might bias the findings. the Teichmann group recently published a Nature Methods paper comparing platforms, which might be helpful.”

Locations of manuscript changes: Supplementary Fig. 22 and lines 242-249, 276-280.

Our usage of the Fluidigm system was based on its availability at our institution. We have added a discussion of the merits of this approach compared to other scRNA-Seq approaches in our Results section titled “Single-cell RNA-Seq identifies resistance phenotypes,” with reference to the Teichmann paper, which we thank the reviewer for sharing with us. We have also added additional data addressing possible batch effects associated with running samples on different Fluidigm chips; please see response to R2-3.

R2-2: “It is well known that single cell RNAseq has high dropout rates (up to 80%). how did the authors deal with this issue?”

Locations of manuscript changes: Supplementary Figs. 19, 21, 22 and lines 242-249, 252-255, 273-280.

We addressed dropout in our original manuscript with pathway analysis, using ssGSEA for publicly available (Molecular Signatures Database) signatures, and ASSIGN for our custom signatures. This approach, by using multiple genes in a pathway, diminishes the effects of individual gene dropout. We only focused on individual genes after pathway analysis indicated that these genes might be important. In our revised manuscript we have explained our approach more clearly, including a schematic figure (new Supplementary Fig. 19c).

In addition, we have added the following data to our revised manuscript to address gene dropout. These results are described in the Results section titled “Single-cell RNA-Seq identifies resistance phenotypes.”

- *Quality control metrics.* In our revised manuscript we have shown quality control analysis of our scRNA-Seq data (new Supplementary Fig. 19a-b), which show that bulk RNA-Seq expression compared to a matched scRNA-Seq experiment simulating a bulk sample (by averaging gene expression across all cells) show strong positive correlation in gene expression ($r = 0.90$) and detect a similar number of genes (~18,000). Further, very few genes detected in bulk RNA-Seq cannot be detected in the simulated bulk scRNA-Seq sample. This suggests that some gene dropout is biological, as others have shown in that single-cell PCR, the gold standard for single-cell expression analysis, reveals less technical dropout in scRNA-Seq than expected (please see doi:10.1038/nmeth.2694). We also show that the number of genes we detect in our scRNA-Seq data is comparable to other studies (Supplementary Fig. 19a).
- *Validation of ssGSEA pathway results using a probabilistic over-dispersion-based approach.* We have analyzed our scRNA-Seq data for each patient using the PAGODA package in R, which utilizes a probabilistic over-dispersion-based approach to detect gene expression and pathway

variation in noisy scRNA-Seq data (new Supplementary Fig. 21). PAGODA was designed specifically with scRNA-Seq data in mind due to its tendency for technical dropout and other biases, and results obtained from this secondary approach strongly corroborate our original findings.

- *Biological replicate analysis.* As shown in more detail in response to R2-3, the same patient sample run on different days on different chips shows very little between-replicate gene dropout (i.e. genes present in one biological replicate and not the other; see new Supplementary Fig. 22b) and strong positive correlation ($r = 0.91$). Thus gene dropout is likely somewhat systematic and does not differ greatly between batches, enabling comparison of noisy genes between samples run, by necessity, on different single-cell capture chips. Further, our statistical approach is to compare the proportion of cells expressing a particular gene between pre- and post-treatment samples (Fig. 5d, and Supplementary Fig. 27, which have not changed in this revised version). This approach, given our biological replicate experiment just described, is appropriate given that technical dropout between pre- and post-treatment samples is likely to be similar, thus enabling strong biological signals to be detectable.

R2-3: “Similarly, batch effects in single cell RNAseq are large and introduce artifacts that can render analyses completely misleading. this issue is not mentioned and apparently has not been addressed when analyzing the samples.”

Locations of manuscript changes: Supplementary Fig. 22 and lines 276-280.

As described above, in our revised manuscript we show data in which we ran scRNA-Seq on replicate patient samples (patient #2 pre-treatment sample) twice on different days (different chips), which we term replicate 1 and replicate 2. Both replicates (reflecting batches) showed the same changes in relevant pathways discussed in our manuscript (EMT, RTK, and TNF pathways; see new Supplementary Fig. 22a) relative to the post-treatment sample. These data indicate that our most statistically significant findings between pre- and post-treatment samples are unlikely to be explained by batch effects. Further, we observed a strong positive correlation between gene expression in the replicate 1 and replicate 2 scRNA-Seq experiments ($r = 0.91$; Supplementary Fig. 22b). We describe these results in our revised manuscript in the Results section titled “Single-cell RNA-Seq identifies resistance phenotypes.”

Thus batch effects, while present, likely cannot explain the most statistically significant differences. Further, we have validated many of our findings using orthogonal approaches, such as biochemical analysis (Supplementary Figs. 23-26, which were in our original manuscript).

R2-4: “The CNA calling algorithm the authors used to analyze their single cell RNAseq data (from Aviv Regev's lab, the melanoma science paper) is very sensitive to the choice of window sizes. The current paper neither describes which window size was used (Im guessing the default, 100), nor were sensitivity analyses performed to study the effects of this choice. It is a randomly chosen parameter that very significantly determines the results so needs to be handled with care.”

Locations of manuscript changes: Supplementary Fig. 28 and lines 304-307.

Thank you for this insightful comment. We used a window size of 101 genes (for each gene we averaged 50 genes upstream, 50 downstream, and the gene itself). In our revised manuscript we have studied the effects of different window sizes (new Supplementary Fig. 28). Smaller window sizes led to unduly high noise, making identification of true CNA regions difficult. Larger window sizes performed similarly to a 101-window size but led to excessive removal of chromosome-terminal regions (due to requiring larger window “priming” regions at the beginnings and ends of chromosomes), which sometimes contained CNAs of interest. Thus the 101-gene window appears suitable for our purposes. These findings are described in the Results section titled “Single-cell RNA-Seq identifies resistance phenotypes.”

R2-5: “Related to the above comment, there might be better ways to identify tumor cells apart from using a quite crude CNA caller together with clustering. For instance, specific driver mutations could be used.”

Locations of manuscript changes: Supplementary Figs. 16, 30 and lines 320-326 (307-318 also).

Again, thank you for this comment and your help. As partly described in response to R1-11, in our revised manuscript we have performed a more comprehensive and stringent mutation detection analysis to detect somatic SNVs in scRNA-Seq reads. To do this, we used the UNCEqR tool, which searches for DNA sequencing-identified mutations in RNA-Seq data, on all patients. This analysis supersedes our previous analysis, which was only performed on patient #1 using a semi-manual approach on only certain hand-picked subclone-specific mutations.

We used point mutations identified in scRNA-Seq as a validation of our single-cell subclone assignments determined from scRNA-Seq inferred copy. At least 10-read coverage was required in scRNA-Seq data at an SNV site in order to perform mutation calling, as well as coverage in at least one cell from each subclone (per inferred scRNA-Seq copy) in the specimen, in order to test for the presence of the mutation in at least one representative cell from each subclone. Further, we only analyzed subclonal SNVs, as truncal SNVs do not aid in assigning cells to specific subclones. Due to high 3' bias of scRNA-Seq read coverage, likely due to the method of poly(A) based library preparation, very few SNVs met these criteria (new Supplementary Fig. 30 for candidates). Only two SNVs in patient #3 met these criteria, which were in 3' UTR regions, and which defined mutually exclusive subclones in patient #3. These mutations could only be found in cells expected to harbor them based on scRNA-Seq inferred copy (Supplementary Fig. 16, bottom). Thus mutation analysis in scRNA-Seq data can be used to validate scRNA-Seq inferred copy but is too sparse to define subclones alone. Our findings are also consistent with sparse mutation detection in scRNA-Seq data by others (doi:10.1038/ng.3806). These findings are described in the Results section titled “Single-cell RNA-Seq identifies resistance phenotypes.”

Also, as described in response to R1-11, we have increased the stringency for assigning single cells to subclones based on scRNA-Seq inferred copy, requiring two CNAs to assign cells to a subclone. We also note the strong corroboration between scRNA-Seq inferred copy and bulk CNA analysis, and the strong co-occurrence of multiple CNAs in the same cell—CNAs one would expect to co-occur based on bulk CNA analysis (see new Supplementary Fig. 29 and updated Supplementary Figs. 14, 16, and 17). These results validate the usage of scRNA-Seq inferred copy to determine the subclone to which single cells belong.

R2-6: “The authors state that they identify point mutations in single cell RNAseq data by aligning with mismatches and keeping a mutation if found in two cells. these are also arbitrary choices that will determine outcomes. please carefully study the effects of these.”

Locations of manuscript changes: Supplementary Figs. 16, 30 and lines 320-326.

As noted above, we have greatly increased the stringency for calling mutations from scRNA-Seq data. The two mutations described in R2-5, which were the only ones with at least 10 read coverage in at least one cell from each subclone (determined by scRNA-Seq inferred copy), had median coverage of 215 and 91. We required a variant allele frequency of at least 0.1 to call a cell as positive for these mutations, which would be a median of ~22 or ~9 mutant reads. These criteria are significantly more stringent than our original scRNA-Seq mutation analysis. As also noted above, these mutations corroborated subclone assignments from scRNA-Seq inferred copy analysis, though even these mutations were too sparse to assign subclones without the aid of scRNA-Seq inferred copy.

R2-7: “The discussion of adaptive therapy needs to be handled with more care. the authors cite the work by gatenby et al but this is a controversial idea, and also not the only method that identifies treatments based on systems approaches (see methods by chmielecki et al, Science Translational Medicine 2011 and others). please present an inclusive discussion of approaches.”

Locations of manuscript changes: Lines 399-417.

This is helpful feedback. In our revised manuscript Discussion section, we have included a discussion of the studies referenced by the reviewer, as well as other approaches for targeting drug-resistant cancers. These approaches include intermittent or other modified dosing schedules, maintenance of the index treatment in combination with an additional therapy, and up-front treatment with resistance-targeting drugs. This additional discussion comprises two paragraphs citing more than 10 relevant studies.

Reviewer #3 Comments

R3-1: *“The relationship between clonal mutational signatures and clonal fitness could be discussed in more depth, with reference to existing literature on the subject of cancer evolutionary dynamics, which is extensive. In particular, it is not clear what is meant for “clonal fitness capacity” (line 210).”*

Locations of manuscript changes: Fig. 4b and lines 224-229.

Thank you for this comment; we agree that this was unclear. In our revised manuscript, we have removed this statement and replaced it with a more detailed discussion of how the presence of different mutational signatures may affect subclonal evolution. We have also cited other studies showing that some mutational processes, such as the APOBEC signature, appear later in lung cancer evolution. Further, we have performed additional mutational signature analysis (new Fig. 4b) and identified mutation signatures that are continuous through evolution and those that tend to change (such as the APOBEC signature) in our patients. These points of discussion and new results are described in the Results section titled “Subclonal heterogeneity and evolution of four breast cancers.”

R3-2: *“Cancer clone phenotypes are evaluated functionally in vitro (lines 319 and on), in terms of their behavior in response to various drug. However, in these assays the potency of the drug is measured, not the proliferative capacity of cells in the presence of those drugs. The authors should discuss this point with reference to recent publications highlighting instead the use of proliferation rate metrics ([1] Nat Methods. 13 (2016) 521–527, and 497-500). Such metrics would be quite appropriate for attempts to predict dynamics of clonal expansion, and possibly time to relapse.”*

Locations of manuscript changes: Supplementary Fig. 33 and lines 371-379.

We concur that IC50-based measurements can be confounded by differing proliferation rates in cells, as described in the references indicated. In the assay shown in Fig. 7a and b, we had also performed a “day 0” (day the drug was added) baseline CellTiter-Glo measurement, in addition to the day 3 measurement performed after 3 days of drug treatment, which was not shown in the original manuscript. The day 0 measurement was done to determine the health and growth of the cells during the assay period and to determine whether any of the drug doses brought cells below the baseline cell number. We used these data to determine proliferation rates in the presence of each dose of drug in this assay (new Supplementary Fig. 33). These data show that the post-treatment patient cells decreased in cell number at the highest dose of combined MEK and Akt inhibitor, indicating the potential for true clinical response, and the second-from-highest dose was cytostatic. In contrast, the pre-treatment patient cells proliferated unhindered at all doses of the drug. These results are described in the Results section titled “Targeting evolving phenotypes using adaptive treatment.”

R3-3: *“A minor technical point: the role and/or necessity for feeder cells in the drug assays should be explained.”*

Locations of manuscript changes: Lines 361-362.

In our revised manuscript we’ve added a statement that the fibroblast feeder system “has been shown to promote the growth of primary cancer cultures, even from small tissue samples, compared to other methods”

and have referenced an additional paper in which the authors grew primary cancer cultures for drug assays using the fibroblast feeder system. This can be found in the Results section titled “Targeting evolving phenotypes using adaptive treatment.”

We again thank the reviewers for their time and contributions to our manuscript.

REVIEWERS' COMMENTS:

Reviewer #1 (Remarks to the Author):

The revised manuscript by Brady et al is much improved, and in my opinion all reviewer comments have been answered satisfactorily. As stated before, I find this a quite impressive study and a genuine tour de force. I'm happy to recommend publication in Nature Communications.

I found two minor issues that I'd like to bring to the authors attention (but these would only result in minor changes and would not require re-review):

1. in patient 4: are the two independent subclones at ~50 really the only possible interpretation here? Could it not also be that there was one subclone comprised of the grey, green and blue mutations at 50%, with the population of grey+green mutations being rare (or: non-sampled!); one cell from this latter ('cryptic') population then evolved further leading to the other observed subclones. This does not really alter their story, but to me seems a more likely explanation than two approx. equal subclones.

2. The direct BRCA2 reversion theory in patient 4, SC12 is not really credible: a base would need to be inserted at exactly that position (out of very many possibilities to get the protein back in-frame), and moreover, that base would have to be a guanine. It may be possible that something else is going on (e.g. a deletion taking out the entire exon, which would also affect the allele frequency), but I don't buy the direct reversion theory.

Reviewer #3 (Remarks to the Author):

I am satisfied with the response to my three concerns. Specifically, there is now a more scholarly discussion of clonal evolutionary dynamics with appropriate references. In a new supplementary figure (fig. 33), clonal proliferation rates are indicated and related to observed clonal expansion or disappearance.

The technical point on feeder layers is also resolved.

Reviewer #1 (Remarks to the Author):

The revised manuscript by Brady et al is much improved, and in my opinion all reviewer comments have been answered satisfactorily. As stated before, I find this a quite impressive study and a genuine tour de force. I'm happy to recommend publication in Nature Communications.

- **Response:** We are very appreciative of reviewer #1's positive comments and suggestions from this and previous rounds of review. They have greatly strengthened the rigor and clarity of our findings, and we appreciate the time taken to thoroughly review our manuscript.

I found two minor issues that I'd like to bring to the authors attention (but these would only result in minor changes and would not require re-review):

1. in patient 4: are the two independent subclones at ~50 really the only possible interpretation here? Could it not also be that there was one subclone comprised of the grey, green and blue mutations at 50%, with the population of grey+green mutations being rare (or: non-sampled!); one cell from this latter ('cryptic') population then evolved further leading to the other observed subclones. This does not really alter their story, but to me seems a more likely explanation than two approx. equal subclones.

- **Response:** This is a very astute observation from reviewer #1, and the alternative subclone scheme suggested by reviewer #1 is indeed a possibility we had considered. We have included an additional Supplementary Figure (Supplementary Figure 12) showing why the scheme shown in Fig. 2d seems more plausible to us. As seen in Supplementary Figure 12, the density analysis at day 0 of mutations in the green mutation cluster (day 0 surviving subclone) shows a large peak at VAF 17.9%, but also a potential second "hump" peak at VAF ~10.0%. All of these mutations became clonal (present in all

cells at VAF ~50%) in subsequent timepoints, indicating that the hump peak represents a subset of the green mutation cluster cells that survived. Thus, this leaves only 7.9% of “VAF space” (17.9% - 10.0%) for the blue mutation cluster to be present, if it is in the same cells as the green large peak. However, the day 0 blue cluster (disappearing subclone) density analysis shows a large peak at VAF of 16.3% (with no minor peaks)—much higher than 7.9%—making the co-occurrence of blue mutations and green mutations in the same cells unlikely (given that blue disappear and green stay and increase to VAF ~50%).

In addition, expert pathological examination revealed that the day 0 sample (a formalin-fixed DCIS sample) was approximately 80% tumor. The scheme we suggest in Fig. 2d would indicate a tumor purity of 68.4%—reasonably close to the pathological estimate of 80%. (The same section that was examined pathologically was the section from which we isolated FFPE scrolls for DNA isolation.) The alternative scheme suggested by reviewer #1 would suggest a tumor purity of approximately 35.8% (2 x 17.9%), which would be less consistent with pathological analysis.

2. The direct BRCA2 reversion theory in patient 4, SC12 is not really credible: a base would need to be inserted at exactly that position (out of- very many possibilities to get the protein back in-frame), and moreover, that base would have to be a guanine. It may be possible that something else is going on (e.g. a deletion taking out the entire exon, which would also affect the allele frequency), but I don't buy the direct reversion theory.

- **Response:** Direct *BRCA2* reversions do indeed seem less plausible than other reversion types. However, in one study of recurrent post-cisplatin *BRCA1/2*-revertant ovarian cancers, 9/13 patients had direct reversion to wild-type, while the remaining four patients had other events such as nearby indels restoring frame (Norquist et al., *Journal of Clinical Oncology* 29:3008-3015; <http://ascopubs.org/doi/abs/10.1200/jco.2010.34.2980>). Among the 9 patients with direct reversions, several had *BRCA1/2* one- or two-nucleotide frameshift deletions (as with our patient) that were restored to wild-type after platinum treatment. We note that in response to this comment, we have cited this paper in our revised manuscript to show direct reversions happen in other situations, and further softened the already tentative language suggesting a direct reversion in SC12.

Further, alternative explanations of our findings, such as deletion of the exon containing the germline inactivating mutation, would not result in increased wild-type reads. (We do in fact see more wild-type reads in the potential revertant subclone in question, SC12 (Supplementary Figure 14).)

Although direct reversion seems unlikely in principle, from the perspective of natural selection direct reversions may confer enhanced fitness, since other reversion schemes include aberrant amino acids (or aberrantly deleted amino acids), which may negatively affect protein folding, function, and stability. This may partly explain why SC12 out-competed revertant SC11, since SC11's reversion included several aberrant amino acids before frame restoration, potentially not allowing full protein function. Further, in late-stage cancer cases, as in this patient, there are easily 10 billion cancer cells in the patient's body (Vogelstein et al., *Nature Medicine* 10:789-799; <https://www.nature.com/nm/journal/v10/n8/full/nm1087.html>). With a conservative mutation rate of 2 mutations per cell division (Tomasetti et al., *PNAS* 110:1999-2004; <http://www.pnas.org/content/110/6/1999.full>), the most recent cell division caused 5 billion x 2 = 10 billion unique mutations. Since the human genome has 3 billion base pairs, this is enough mutations to cover more than every position in the genome in at least 1 cell—and that is only accounting for the most recent cell division. Thus, patients with systemic disease have nearly limitless

heterogeneity on which selection may act, and in this setting the possibility of at least one cell with a direct reversion appears more plausible.

Reviewer #3 (Remarks to the Author):

I am satisfied with the response to my three concerns. Specifically, there is now a more scholarly discussion of clonal evolutionary dynamics with appropriate references. In a new supplementary figure (fig. 33), clonal proliferation rates are indicated and related to observed clonal expansion or disappearance. The technical point on feeder layers is also resolved.

- **Response:** We very much appreciate reviewer #3's positive comments and suggestions. They have improved the clarity of our manuscript and strengthened our drug-response assay data. We thank reviewer #3 for the time taken to review our manuscript.